



# On the laminar-turbulent transition mechanism on Multi-Megawatt wind turbine blades operating in atmospheric flow

Brandon Arthur Lobo[1], Özge Sinem Özçakmak[2], Helge Aagaard Madsen[2], Alois Peter Schaffarczyk[1], Michael Breuer[3], and Niels N. Sørensen[2]

[1]Mechanical Engineering Department, Kiel University of Applied Sciences, D-24149 Kiel, Germany
[2]Department of Wind Energy, Denmark Technical University, DK-4000 Roskilde, Denmark
[3]Professur für Strömungsmechanik, Helmut-Schmidt-Universität Hamburg, D-22043 Hamburg, Germany
**Correspondence:** Brandon Arthur Lobo (brandon.a.lobo@fh-kiel.de)

**Abstract.** Among a few field experiments on wind turbines for analyzing laminar-turbulent boundary layer transition, the results obtained from the DAN-AERO and Aerodynamic Glove projects provide significant findings. The effect of inflow turbulence on the boundary layer transition and the possible transition mechanisms on wind turbine blades are discussed and compared to CFD simulations of increasing fidelity (RANS, URANS and LES). From the experiments, it is found that the transition scenario changes even over a single revolution with bypass transition taking place under the influence of enhanced upstream turbulence from e.g. wakes and natural transition at other times under relatively lower inflow turbulence conditions. This change from bypass to natural transition takes place at azimuthal angles directly outside the influence of the wake indicating a quick boundary layer recovery. The importance of a suitable choice of the amplification factor to be used within the $e^N$ method of transition detection is evident from both the RANS and URANS simulations. The URANS simulations which simultaneously check for natural and bypass transition match very well with the experiment. The LES predictions with anisotropic inflow turbulence show the shear-sheltering effect and a good agreement between the power spectral density plots from the experiment and simulation is seen. A condition to easily distinguish the point of transition to turbulence based on the Reynolds shear stress term is also observed. Overall, useful insights of the flow phenomena are obtained and a remarkably consistent set of conclusions can be drawn.

## 1 Introduction

Wind turbines operate continuously in the atmospheric boundary layer and often under augmented inflow turbulence conditions such as in the wake of other wind turbines, on irregular terrains, in extreme weather conditions and in wind shear. However, the airfoil sections used for the blade designs are tested in wind tunnel experiments that do not account for the unsteady inflow turbulence and the rotation of the rotor. This causes deviations of the aerodynamic performance of the wind turbine operating in atmospheric conditions from the predicted performance when these conditions are not properly taken into account. A common procedure in blade designs to account for the uncertain operating conditions on a rotor is to use an empirical blend of airfoil data based on free and fully turbulent boundary layer flow. Moreover, the turbulent kinetic energy (TKE) spectrum in the free atmosphere is remarkably different from that typically generated in a wind tunnel. This was clearly identified in the DAN-



AERO experiments (Madsen et al., 2010b) carried out in 2009, where the inflow to an airfoil section in the wind tunnel and on a 2.0 MW rotor was compared characterized by the pressure fluctuations in the laminar boundary layer at the leading edge of the airfoil section. Much higher energy contents of the pressure fluctuations from the inflow were seen on the blade section on the rotor in a frequency interval up to about 300-500 Hz. Adding a turbulence grid in the wind tunnel decreased the difference somewhat, but then caused the spectrum in the wind tunnel to exceed the rotor spectrum for frequencies above 500 Hz.

In Schaffarczyk et al. (2017) it is reported that in the case of atmospheric flow, the TKE spectrum possesses a maximum at around 0.01 Hz followed by a decrease according to Kolmogorov's $k^{-5/3}$ law, whereas in a wind tunnel experiment with a turbulence grid much more energy is distributed in the kHz range.

As a part of the MexNext project (Boorsma et al., 2018) transitional studies were carried out on a rotor model in a wind tunnel under varying operating conditions, but in the absence of added inflow turbulence. An analysis of the experimental data showed that natural transition of the Tollmien-Schlichting (T-S) kind took place (Lobo et al., 2018). As a part of this project, the first comparisons between CFD predictions using turbulence models including transition modeling and such an experiment were conducted by four groups and a reasonable agreement with the transition location compared to the experimental data was found (Schaffarczyk et al., 2018). However, in the case of atmospheric inflow, different transition scenarios are expected. This depends on, for example, the length scales and turbulence intensity ($T.I.$) of the flow field. It is well known that for a $T.I.$ in the order of $0.5$ to $1\%$ transition through TS instabilities can be expected (Reshotko, 1976, 2001) while with a rise in $T.I.$ bypass transition (Morkovin, 1969) occurs. Bypass transition is a broad term encompassing transition scenarios, where the initial growth is not described by the eigenmodes of the Orr-Sommerfeld equation. In such scenarios, transient growth is often observed, where decaying perturbations in terms of the eigenfunctions of the Orr-Sommerfeld equation undergo temporary amplifications on account of their non-orthogonality (Butler and Farrell, 1992; Reshotko, 2001). This can result in streaks in the streamwise direction, which are regions of relatively high or low velocity relative to the mean flow. Such characteristics were observed in the measured surface pressure fluctuations in the DAN-AERO experiments (Madsen et al., 2010b).

This non-modal growth can be related to the free-stream disturbances through a mechanism known as shear sheltering (Hunt and Carruthers, 1990), which permits disturbances to penetrate up to a certain depth within the boundary layer. The penetration depth was found to depend on the frequency of the disturbance and the flow Reynolds number with lower frequencies being able to penetrate deeper into the boundary layer (Jacobs and Durbin, 1998). Zaki (2013) illustrated the shear filtering effect with a model problem that used two scales: the lower frequency penetrated the boundary layer near the leading edge resulting in the generation of streaks as a boundary layer response. This is followed by their breakdown through a process known as the lift-up mechanism (Kline et al., 1967), where negative perturbation streaks rise within the boundary layer. The external high-frequency disturbances, which are limited in their penetration, provide excitation for the growth of an outer instability leading to turbulent breakdown of the streak.

To better understand the transition phenomena on a wind turbine in the field, experimental data are necessary. A limited number of field experiments dedicated to laminar-turbulent boundary layer transition on wind turbine blades in atmospheric conditions is available as listed in Table 1. The analysis of the DAN-AERO and Aerodynamic Glove[1] experiments release

---

[1]The term *aerodynamic glove* was borrowed from Seitz and Horstmann (2006).





significant amount of information on the transition mechanisms on wind turbine blades. This paper focuses on the results
from these experiments in comparison with numerical simulations with models of different fidelity. First, the experimental

findings from the DAN-AERO and Aerodynamic Glove projects focused on laminar-turbulent boundary layer are presented.
Furthermore, numerical analysis for boundary layer transition that were conducted for IEA Wind Task 29 are summarized. New
numerical analysis results are presented. Results from the DTU in-house CFD solver EllipSys3D (Sørensen, 1995; Michelsen,
1992, 1994) URANS (Unsteady Reynolds-Averaged Navier-Stokes) simulations in comparison to DAN-AERO experiments
and LES (Large-Eddy Simulations) in comparison with Aerodynamic Glove Experiments are discussed in detail. The LES

predictions rely on the HSU in-house code $\mathcal{LESOCC}$ (Breuer, 1998, 2000, 2002, 2018) combined with the synthetic inflow
generator by Klein et al. (2003). Finally, general findings from these unique field experiments and simulations are summarized
to shed light on different transition mechanisms and the effect of the atmospheric flow in transition on wind turbines.

**Table 1.** Field experiments on boundary layer transition on wind turbine blades (regenerated from Özçakmak (2020)).

| Project | HAT25 | DAN-AERO | Aerodynamic Glove | Aerodynamic Glove (MM92) |
|---|---|---|---|---|
| Year | 1983 | 2009 | 2011 | 2018 |
| Experimental technique | microphones glued on surface | flush-mounted high-frequency microphones | hot-film and pressure tubes | microphones and thermographic cameras |
| Blade length (m) | 12.05 | 38.8 | 15 | 45.3 |
| Reynolds number ($\cdot 10^6$) | 1.5–3.5 | 3–5.2 | 1–2.5 | 1–5 |
| Reference | Van Ingen and Schepers (2012) | Madsen et al. (2010a) Troldborg et al. (2013) | Schaffarczyk et al. (2017) Schwab et al. (2014) | Reichstein et al. (2019) |

## 2    Experimental findings

This section discusses the experimental findings from the projects DAN-AERO and Aerodynamic Glove on boundary layer

transition analysis on wind turbines.

### 2.1    DAN-AERO

A significant part of the DAN-AERO project (Madsen et al., 2010a; Troldborg et al., 2013) was concentrated on the investi-
gations of the laminar-turbulent boundary layer transition on LM-38.8 blade of the 2 MW NM-80 rotor placed in a wind farm
and on a 2D airfoil, which was a replicate of this blade section. The blade section, which is 36.8 meters from the hub, was

equipped with high-frequency flush-mounted microphones and pressure taps. Detailed boundary layer transition analyses were





performed in subsequent studies (Özçakmak et al., 2019; Özçakmak et al., 2020) by analyzing the DAN-AERO data further
and conducting CFD simulations for the 2D and the full rotor flow.

### 2.1.1 Experimental findings from DAN-AERO

Boundary layer transition was detected based on the surface pressure fluctuations captured by the high-frequency microphones.
Power spectral densities (PSD) of these fluctuations are integrated in a frequency range from 2 kHz to 7 kHz to calculate
root-mean-square (RMS) values was chosen according to a parametric study for various Reynolds numbers, where higher
frequencies are excluded due to resonance in the tubing system caused by the pinhole placement of the microphones and the
lower frequencies are eliminated due to the possible effect from inflow turbulence or large eddies that would interfere the
spectra. On condition that a certain threshold is exceeded, a sudden chordwise increase in RMS values indicates the transition
location.

It is found that it is possible to determine the type of transition from the spectra. Different transition mechanisms were
observed in the experiments. When it is a natural transition, the flow goes through a gradual process of three steps. In the
receptivity step, initial disturbances occur in the boundary layer due to outside disturbances. Disturbances such as Tollmien-
Schlichting (T-S) waves are triggered inside the boundary layer. Then, unstable waves go through the second step of linear
instability by a linear amplification process (Arnal et al., 1998). In the third step, non-linear interactions in the form of secondary
instabilities occur when the unstable wave amplitude grows and reaches a finite value (Reed and Saric, 1989). It is possible to
follow T-S wave frequencies with a sufficient amount of microphones placed chordwise. Figure 1 shows PSD vs. frequency $f$
at different chordwise positions ($x/c$) on the pressure side of an airfoil model tested in a wind tunnel at $Re = 3 \cdot 10^6$, AOA=
$0°$. This model is a replication of the LM-38.8 blade section. The small peaks that are observed at the laminar spectra are due
to the blade passing frequency of the fan in the wind tunnel (see Özçakmak et al., 2019). Laminar spectra are observed at the
chord positions until $x/c = 0.48$ (blue line). The microphone at this position shows a transitional flow spectra with a T-S wave
peak. Then the flow becomes turbulent downstream of $x/c = 0.54$ (orange line). Since transition does not take place at a point
but is a process occurring over a certain distance, more resolution of the experimental instruments would show a longer process
in natural transition. In that condition, the mentioned peak grows before reaching a fully turbulent spectrum. It should be noted
that the other peaks in the spectra are due to acoustic resonances in the wind tunnel.

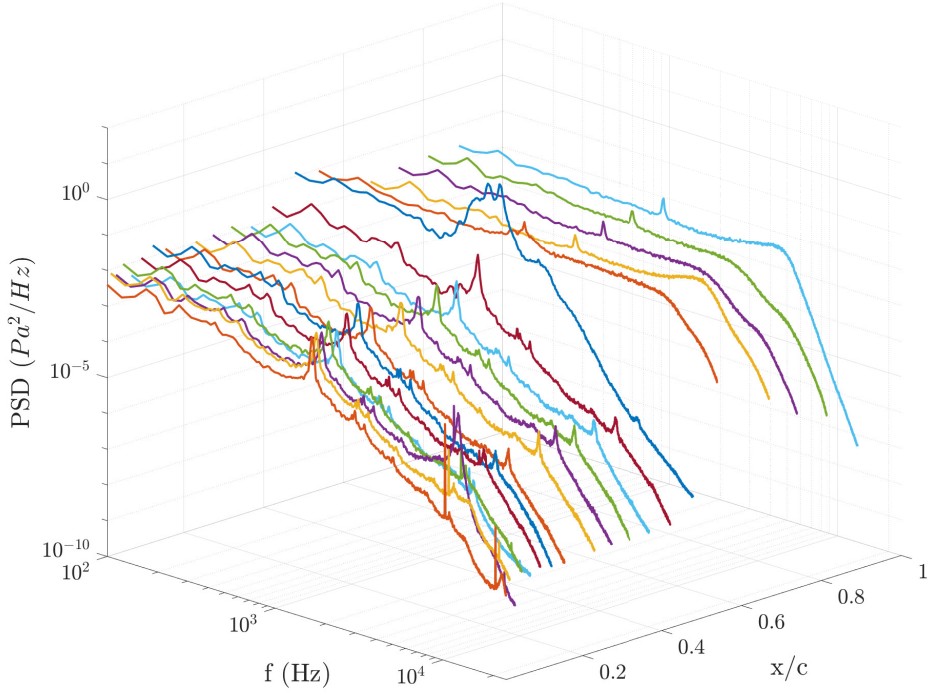

**Figure 1.** Chordwise power spectral density based on the pressure on the lower side of the airfoil, $Re = 3 \cdot 10^6$, AOA= $0°$ (Özçakmak, 2020).

As the analysis moves from the flow on a 2D airfoil to a full-scale wind turbine, inflow turbulence is found to be the predominant factor that influences the transition behavior as well as the angle of attack changes in a single revolution. Also surface roughness has an important influence on the transition characteristics.

Since there was a lack of high frequency velocity fluctuation measurements in the field experiments, pressure fluctuations from

the surface microphones are used to analyze the inflow turbulence characteristics in detail. Therefore, inflow turbulence is interpreted from a leading-edge microphone at $x/c = 0.2\%$ in the laminar boundary layer using pressure fluctuations as shown in Equation (1), where $P_{s,p}$ is the power spectral density of the microphone signal, $\sigma$ is the standard deviation, and $f_1$ and $f_2$ are the integration frequency boundaries.

$$P_{\text{rms}}^2 = \int\limits_{f1}^{f2} P_{s,p}\, df \quad , \qquad \sigma = \sqrt{P_{\text{rms}}^2} \quad . \tag{1}$$

Figure 2 shows the PSD from this microphone for both wind tunnel and field experiments. It is obvious from the wind tunnel experiments that the pressure spectra from the clean tunnel and the tunnel with the turbulence grid represent the differences in the inflow conditions. Thus, this microphone can be used to demonstrate the inflow turbulence characteristics. It is seen that the tunnel with a grid has a higher amplitude of pressure fluctuations compared to the clean tunnel case. Two angles of attack are





presented here which represent the effective range of angles seen in the field measurements. As for the field experiments, the
turbine is operated in two different pitch settings at $1.25°$ and $4.75°$, respectively. For both settings a low and a high turbulence
intensity value (calculated from the inflow velocity data obtained by the Pitot tube on the blade) are presented. Since the
test turbine was placed in a wind farm at certain wind directions, it operates under a wake-affected inflow, mentioned as the
percentage of the rotor disk area with wake-affected inflow. It is seen from the PSD data obtained in field experiments that as
the angle increases the magnitude of the PSD also increases (see red dashed vs. red solid line and yellow dashed vs. yellow
solid line). On the other hand, for the same pitch setting, increased inflow turbulence levels move the PSD magnitude to even
higher values. This increase is mostly visible in the lower frequencies up to approximately 400 Hz. In a previous study, it is
found that integrating the spectra from 100 Hz to 300 Hz ($f_1$ and $f_2$ in Equation (1)) for a high-frequency microphone placed
in proximity for the leading edge,differences in inflow turbulence characteristics between wind tunnel and rotor flow can be
interpreted.

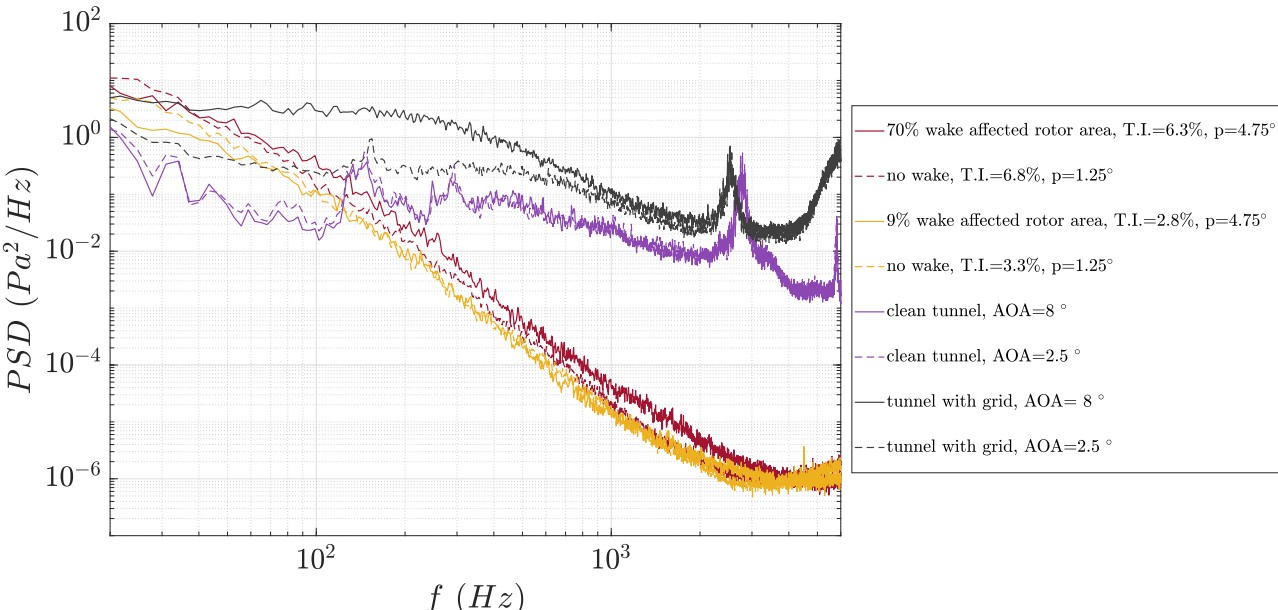

**Figure 2.** Inflow spectra characterized by the pressure fluctuations in the laminar boundary layer for wind tunnel and field experiments
obtained by a microphone placed at $x/c = 0.2\%$ .

Figure 3 demonstrates a case where the inflow velocity is 6.3 m/s and $T.I.$ is 6.3%. The turbine has a pitch setting of $4.75°$
and it is operating in a wake resulting from an upstream turbine, where 69% of the rotor area is affected. A sketch for the
wake-affected rotor area and the azimuthal notation is shown in Figure 4. The various PSD lines are obtained from different
time segments by a microphone placed at $x/c = 13\%$. This microphone, at different times, due to the fact that the angle of





attack and the inflow turbulence changes, operates under laminar, transitional and turbulent flows. At the time of 2.5 s it is under transitional flow, and at 4.1 and 8 s it shows laminar spectra. At 6.4 s it is operating under turbulent conditions.

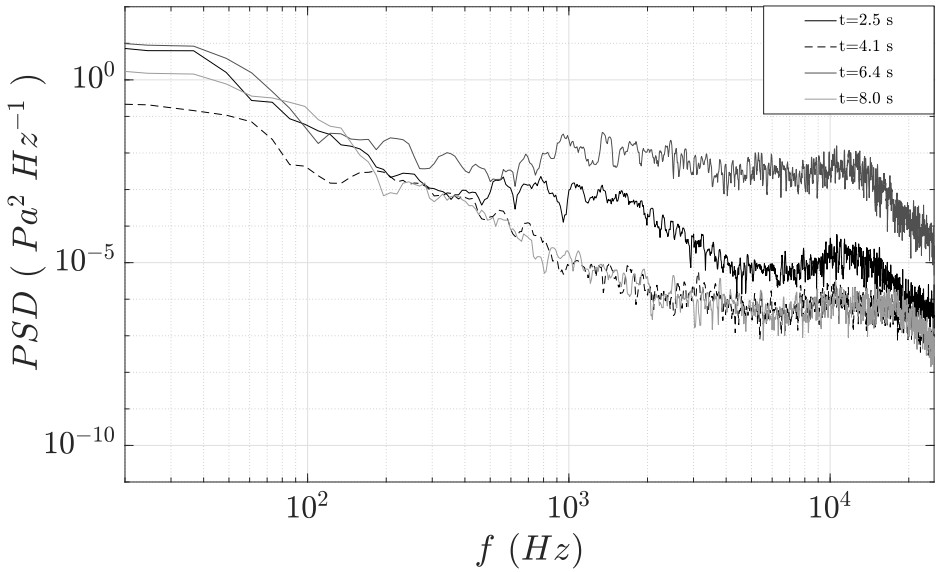

**Figure 3.** Comparison of the PSD from different time indices for the same microphone placed at $x/c = 13\%$.


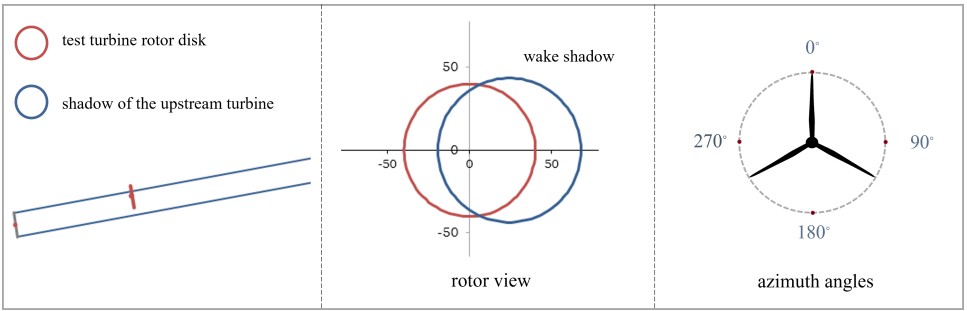

**Figure 4.** Wake-affected rotor area and the azimuthal notation.

This movement in transition locations is also shown in a spectrogram plot (Figure 5) of the chordwise pressure levels, $L_p$ (dB) in time for this case. Sound levels are calculated according to Equation (2), where the reference pressure ($P_{\text{ref}}$) is 20 $\mu$ Pa.

$$L_p = 20 \cdot \log_{10}\left(\frac{P_{\text{rms}}}{P_{\text{ref}}}\right) \tag{2}$$

The sharp color changes means there is a sudden increase in the standard deviations of the pressure fluctuations above a

certain threshold, which indicates transition.

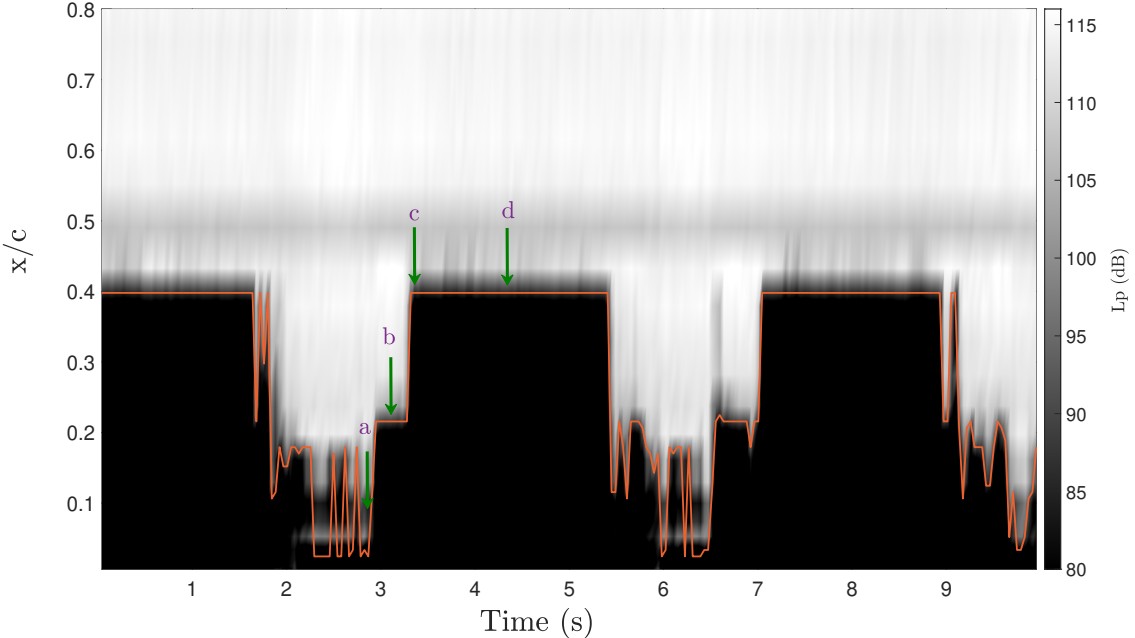

**Figure 5.** Chordwise pressure levels ($L_p$) with a line of detected transition locations. The arrow indicates the time instances where the chordwise PSD values were plotted in Figure 6.

Both a change in the location of the transition due to the angle of attack changes, and also due to inflow turbulence can be seen in a single revolution. The locations where the transition is detected at $x/c = 0.4$ (denoted c and d) and $x/c = 0.22$ (denoted b) are due to the angle of attack changes. However, the locations where the transition moves close to the leading edge are possibly due to the direct effect of inflow turbulence (bypass transition) (denoted a) in addition to the effect of the angle changes.

For further analysis, the chordwise spectra of these time segments are analyzed and presented in Figure 6. It can be seen from Fig. 6- b that, when there is a natural transition, there is a growth of the T-S waves that occurs as a peak in the spectra before the flow becomes turbulent as also demonstrated for the wind tunnel case in Figure 1. The peak starts at $x/c = 19.9\%$ and it grows further downstream until a fully turbulent spectrum is observed. The weighted average of the T-S wave frequency is around 809 Hz. A peak which is slightly above the turbulent spectra is also visible for the case where the transition is found at $x/c = 39\%$ in Fig. 6-c and d. However, at this location, the transition is possibly occurring on a small portion of the chord so that only one microphone was able to catch this peak. There is a more noticeable peak in case c than in case d, possibly because the c position is just the start of the change in the transition location from $21\%$ to $39\%$. At point d, the transition position is already established at $39\%$. For the bypass transition case (Fig. 6-a), it can be sen that the PSD magnitude is much higher in the lower frequency range up to approximately 400 Hz compared to the natural transition cases (b, c and d). In this case, a T-S wave pocket growth is not observed that causes a peak in the spectra; but instead, an increase in the high-frequency



fluctuations of the spectra is found as the flow becomes turbulent. Moreover, all the microphones, even the ones in the laminar range response to this inflow turbulence also at high frequency ranges.

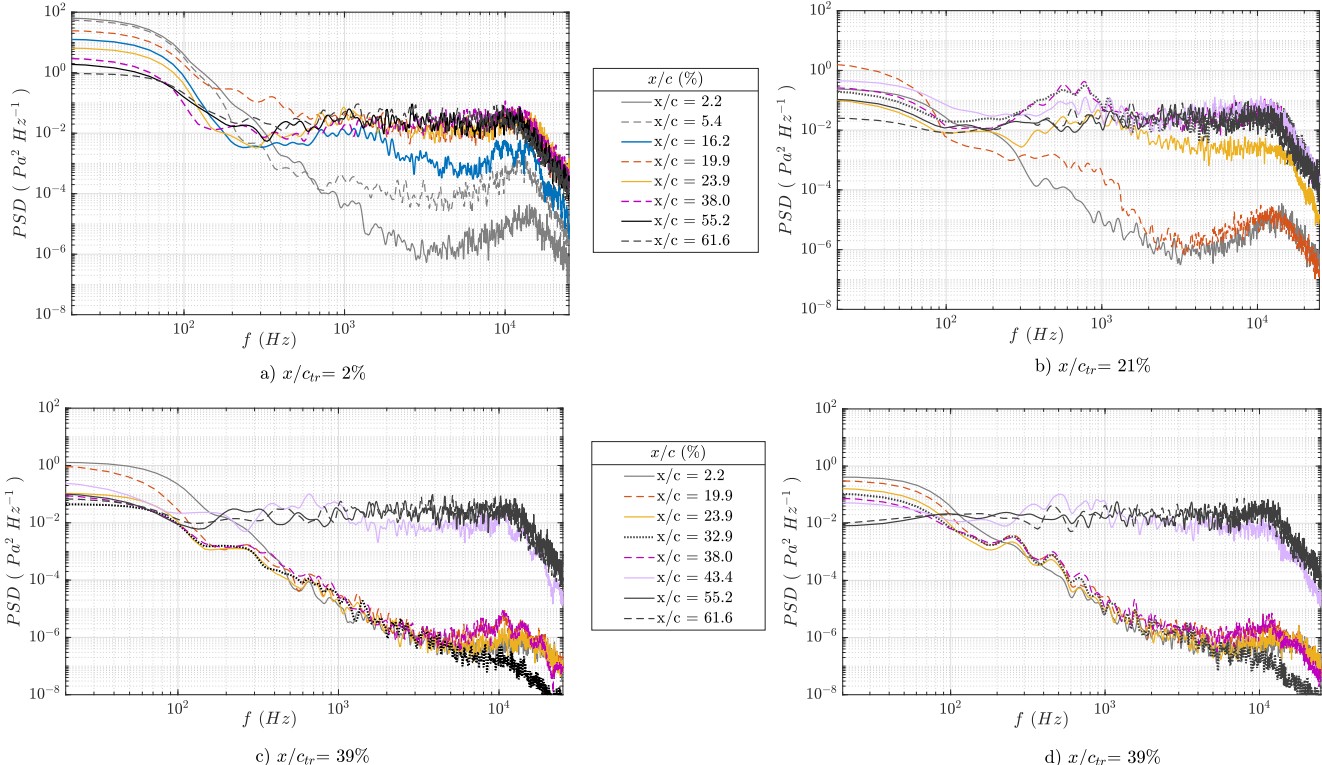

**Figure 6.** PSD at various $x/c$ positions at different time indices (corresponding to the points a, b, c, d shown at Figure 5) during a single revolution where the transition is detected at the chordwise location of a) $x/c = 2\%$, b) $x/c = 21\%$, c) $x/c = 39\%$, and $x/c = 39\%$.





## 2.2  Aerodynamic Glove

After preliminary investigation on a 15 m ENERCON E30 blade, see Schaffarczyk et al. (2017), a similar measurement campaign was undertaken on LM 43P blade mounted on a Senvion (REpower) MM92, see Reichstein et al. (2019) and Fig. 7.

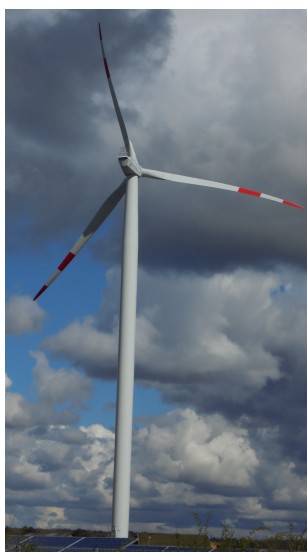

**Figure 7.** Research turbine REpower (SENVION) MM92 at Eggebek, Germany.

Transition was intended to be detected by an array of microphones (like in the DAN-AERO experiment) but now enhanced by two teams (DLR Göttingen and Deutsche Wind Guard Engineering, Bremerhaven) who performed ground-based thermography. Due to the very short time of data compilation (a few hours) and a non-perfect surface preparation or a temporary

fouled surface by dust and/or insects only a few data sets with large areas of assumed laminar boundary layer state could be recorded, see Figs. 8 and 9 for a typical example. The brighter areas in the picture indicate higher temperatures which is interpreted as laminar flow. Magenta colored parts show the pure blade. Nevertheless, the main finding was equal accuracy of microphones and thermography with regard to determination of transition location.

### 2.2.1  Mechanisms of transition

Any conclusion about the type of transition mechanism relies on an underlying theoretical model. In the case of the glove experiments variants of the $e^N$-method, assuming a Tollmien-Schlichting scenario to fully developed turbulence, was applied. Therefore, it seems only to be possible to detect deviations from this specific scenario by deviation of the expected transition location. As far as the observed locations of transition could be compared by CFD enhanced by transition prediction models (for example DLR's FLOWer with prediction module *prdmdl*) no such strong deviation could be found. One difficulty was a

justified choice of the N-factor, often (esp. in wind tunnel measurements) related to the turbulence intensity, see Eq. (3):

$$N(T.I.) = 3.56 - 6.18 \cdot \log_{10}(100 \cdot T.I.) \,. \tag{3}$$



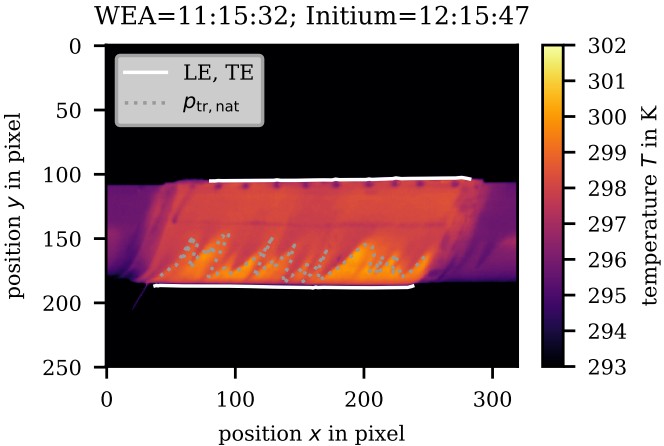

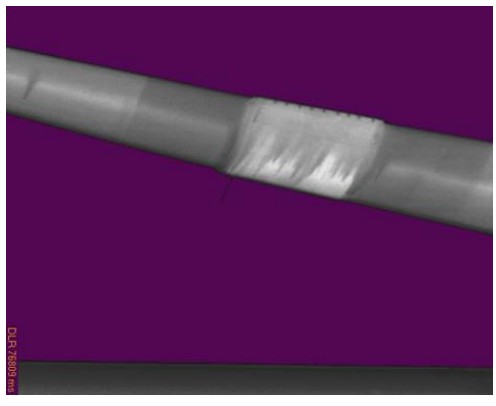

**Figure 8.** Thermographic picture processed by DWGE. Leading edge (LE) bottom. Trailing edge (TE) further up. The dotted line indicates automatically recognized transition by the post-processing system.

**Figure 9.** Thermographic picture by DLR. Leading edge (LE) bottom. Light areas are assumed to be in a non-turbulent (laminar) state. Used by permission of DLR.

## 3 Numerical analysis

To gain further insight into the transition phenomena on wind turbines operating under atmospheric conditions, full rotor simulations and simulations across wind turbine blade cross-sections have been conducted. This section discusses results from
CFD investigations of increasing fidelity with the goal of showcasing the capabilities and contributions of the different methods at their present state of the art. The following simulation methods will be discussed in the subsequent sub-sections:

- RANS predictions with statistical turbulence models including transition modeling mostly as described in more detail in Schepers et al. (2021).

- Unsteady Reynolds-Averaged Navier-Stokes simulations as succeeding work for the model in the DAN-AERO project,

- Large-eddy simulations as a follow up of the Aerodynamic Glove project.

### 3.1 RANS - Findings from IEA Wind TCP Task 29, Subsection 3.6

The analysis presented in this sub-section has been performed as part of the IEA Task 29, Subsection 3.6 (Schepers et al., 2021) on boundary layer transition. It is included here, in part, with the goal of presenting the capabilites of todays' RANS codes with respect to wind turbine transition modeling. As a part of this IEA Task a limited number of teams performed 3D-CFD
predictions including transitional modeling: DTU using EllipSys3D, IAG using FLOWer and KUAS using FLUENT.





**Table 2.** Meshes prepared for CFD models.

| Origin | Extension | Size/M | Blocks | Interface |
|---|---|---|---|---|
| DTU | 360° | 14.1 | 1 | no |
| IAG/DLR | 360° | 29.2 | 3 | Chimera |
| FORwind/IWES | 120° | 24.5 | 3 | AMI |

Presently, the methods typically used for the solution of RANS employs the SIMPLE algorithm for the pressure-velocity coupling together with a Gauss-Seidel algebraic multigrid method and a spatial discretization of the 2nd order (upwind).

On account of the different requirements of the different codes, three different grids (see Table 2) were used by the teams for the purpose of the study. For example, at the Kiel University of Applied Sciences and using the commercial solver FLUENT,
the original DTU mesh did not converge. This was probably due to the high aspect-ratio cells close to the blades wall. The IAG/DLR's mesh uses a Chimera technique with overlapping volumes. Arbitrary mesh interfaces (AMI) couples/interpolates different blocks at close but different 2D-interfaces. As a consequence, although a lot of effort was spent, it was not possible to convert IAG/DLR's Chimera mesh for use with FLUENT. Therefore, the FORwind/IWES mesh prepared by Leo Hönig was employed.

As a turbulence model Fluent uses Menter's SST-k-$\omega$ and to model laminar-to-turbulent transition Menter's $Re_\vartheta - \gamma$ transitional model (NN-, 2013) based on transport equations is applied. It is well known that $e^N$ models are superior for modeling natural transition, see Özlem et al. (2017). TAU uses its transition module (Krimmelbein, 2009) which has implemented the $e^N$ method and DTU's EllipSys3D so far only for the Mexico blade has been reported in Heister (2016). The transition models used by FLOWer and OpenFOAM resp. are described in detail in Schepers et al. (2021).

Figure 10 summarizes the findings for the transition line for computational case 1 of the IEA Task ($v_{in} = 6.1$ m/s and 12.3 rpm). Two groups of near-by results seems to be visible: The first group consists of two applications of Menter's $Re_\vartheta - \gamma$ model and the second one is the Drela's model, and the *FLUENT Bra* case. The difference between FLUENT pre(liminary) and FLUENT Bra(ndon Lobo) is due to a much larger (50 k increased to 200 k) number of iterations despite the fact that most of the residuals (mass flow, for example) seemed to be converged.

Table 3 gives a short overview of the results for global values like $c_T$ and $c_P$. Data from other groups were collected mainly from the presentation at the 2019 meeting at NREL, Boulder, CO, USA. It has to be noted that Troldborg et al. (2013) reported transitional simulations as well. Three groups (LM, DTU and Siemens) were mentioned there. An average of $c_P = 0.482 \pm 0.015$ can be deduced including transition and $c_P = 0.445 \pm 0.005$ for fully turbulent flow (see Chapter 14 of Troldborg et al. (2013) for details). From the BEM description given, a Reynolds number (based on local chord and rotation
speed) can be estimated to lie between 3 and 4 million.

It has to be noted that convergence in terms of residuals is very different from what is recommended as best-practice, see Fig. 11. The residuum for the mass seems to stagnate at a level of about $4 \cdot 10^{-4}$, those for the velocities decrease very slowly down to $10^{-5}$ for the first 38 k iterations



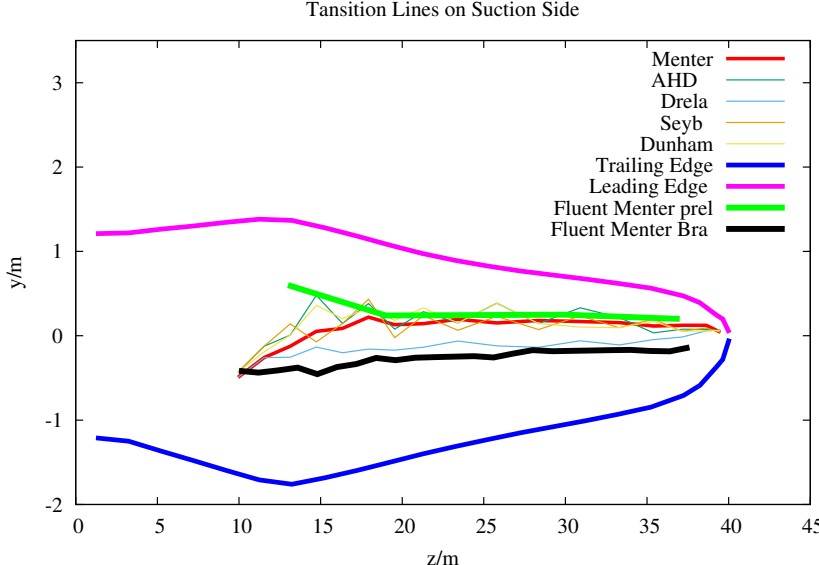

**Figure 10.** Calculated transition lines for an inflow velocity of 6.1 m/s and 12.3 rpm using a variety of laminar-to-turbulent prediction methods. All computations except those labeled *FLUENT* have been performed by G. Bangga, IAG, University of Stuttgart, Germany, now DNV, Bristol, UK. From Schepers et al. (2021).

**Table 3.** Results for $c_T$ and $c_P$ from various computations. wt-perf is an outdated BEM-code from NREL (Buhl, 2004) for aerodynamics (power, thrust, bending moment) only. These values have been incorporated for reasons of comparison only. ft means fully turbulent and tr transitional.

| Solver | Mesh | $c_P$ | $c_T$ |
|---|---|---|---|
| BEM (wt-perf) | (-) | 0.429 | 0.820 |
| EllipSys3D | DTU | 0.483 | 0.824 |
| TAU | IAG/DLR | 0.483 | 0.821 |
| FLOWer | IAG/DLR | 0.471 | 0.820 |
| openFOAM | FORwind/IWES | 0.441 | 0.798 |
| FLUENT ft | FORwind/IWES | 0.417 | 0.763 |
| FLUENT tr | FORwind/IWES | 0.433 | 0.774 |

From Table 3 it is clearly seen that the overall performance prediction with FLUENT is visibly (almost 10%) smaller than the
results of other codes (EllipSys3D, tau and FLOWer). A slight deviation in pitch (of about $0.4°$) was found, but was discarded as the main reason for under-prediction, because BEM-calculations showed that a change of about 1% would result only. So, unfortunately, up to now it remains open what causes this rather big differences.



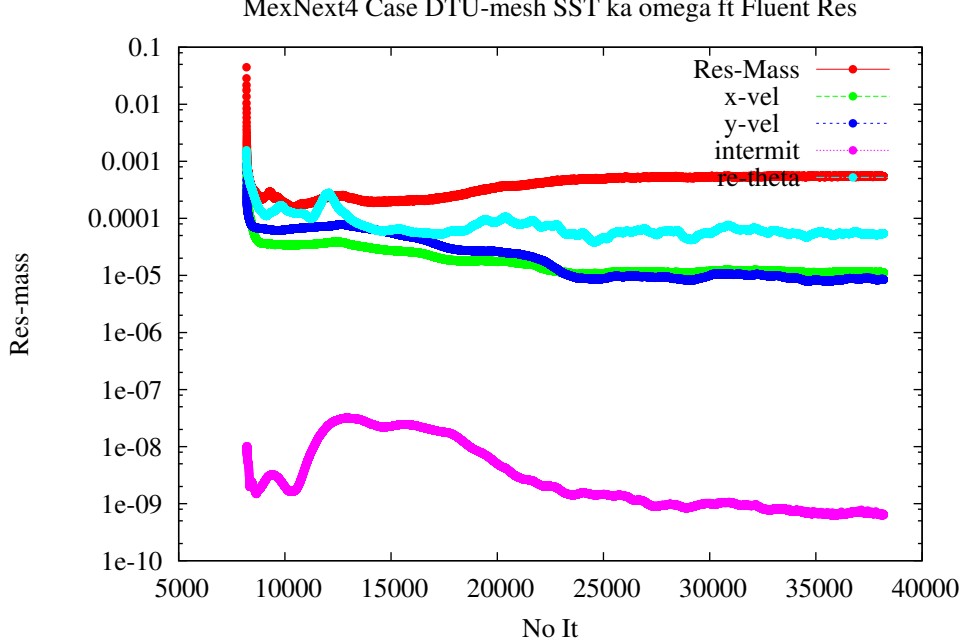

**Figure 11.** Residuals for a typical transitional FLUENT job using FORwind/IWES' mesh and Menter's $Re_\vartheta - \gamma$ transitional model. As the location of the transition line was not smooth after 50 k iterations about 200 k more were added.

## 3.2 URANS full rotor simulations for DAN-AERO experiments

Unsteady Reynolds-Averaged Navier-Stokes simulations are performed using DTU in-house CFD solver EllipSys3D (Sørensen,
1995; Michelsen, 1992, 1994), using the exact model geometry from the DAN-AERO field experiments. The geometry and details of the mesh generation and simulations setup can be found in previous studies (Özçakmak et al., 2020; Özçakmak et al., 2019).

A semi-empirical $e^N$ model (Drela and Giles, 1987) is used to model transition in EllipSys3D. In addition to the natural transition scenario, a bypass criterion can be used together with the $e^N$ model. In order to simulate high turbulence intensity
cases, where bypass transition is observed, the empirical model of Suzen and Huang (2000) is used. In the code, criteria for both the natural and bypass transition are checked simultaneously. For the $e^N$ model, different amplification ratios ($N = 0.15$ to 7) and for the bypass model the relevant turbulence intensity values are introduced as an input.

Simulating all the inflow conditions with various atmospheric turbulence occurrences at once in an URANS setup and at the same time ensuring a detailed transition analysis is a highly complex problem. Therefore, all the occurrences are simulated in
different simulation setups which includes the free-stream velocity range from 5 to 8.5 m/s (measured by the meteorological





mast), and turbulence intensity values from 2.8% to 6.8%. A mesh is generated for the two different pitch angle (1.25° and 4.75°) settings.

Figure 12 shows the intermittency values at the blade section $y = 36.8$ m from the hub obtained from the full rotor simulations for a free-stream velocity $U_\infty$ of 6.3 m/s and the pitch setting of $p = 4.75°$. The intermittency factor $\gamma$ in the transition model

governs the transition to turbulence by controlling the effective viscosity. The intermittency value $\gamma$ of 1 represents turbulent flow and the value 0 represents laminar flow. The transition regions on the upper and lower surfaces of the blade section can be visualized in this way. Detailed information on the intermittency factor and its formulation in the transition model can be found in a previous publication by Özçakmak et al. (2020).

All the visualizations and post-processing is performed using the FieldView software (FieldView, 2017). The four cases

presented here are separate simulations, where the $e^N$ natural transition model is used for various $N$ numbers of 7, 3 and 0.15 as shown in Figs. 12-a, b and c, respectively. Moreover, Fig. 12-d shows the results for the bypass transition model, where the turbulence intensity value is set to 6.3 %. The transition point moves upstream towards the leading edge as the amplification ratio is reduced (i.e., higher turbulence intensity values as shown by the Mack (1977) relation: $N_{tr} = -8.43 - 2.4 \cdot \ln(T.I.)$.

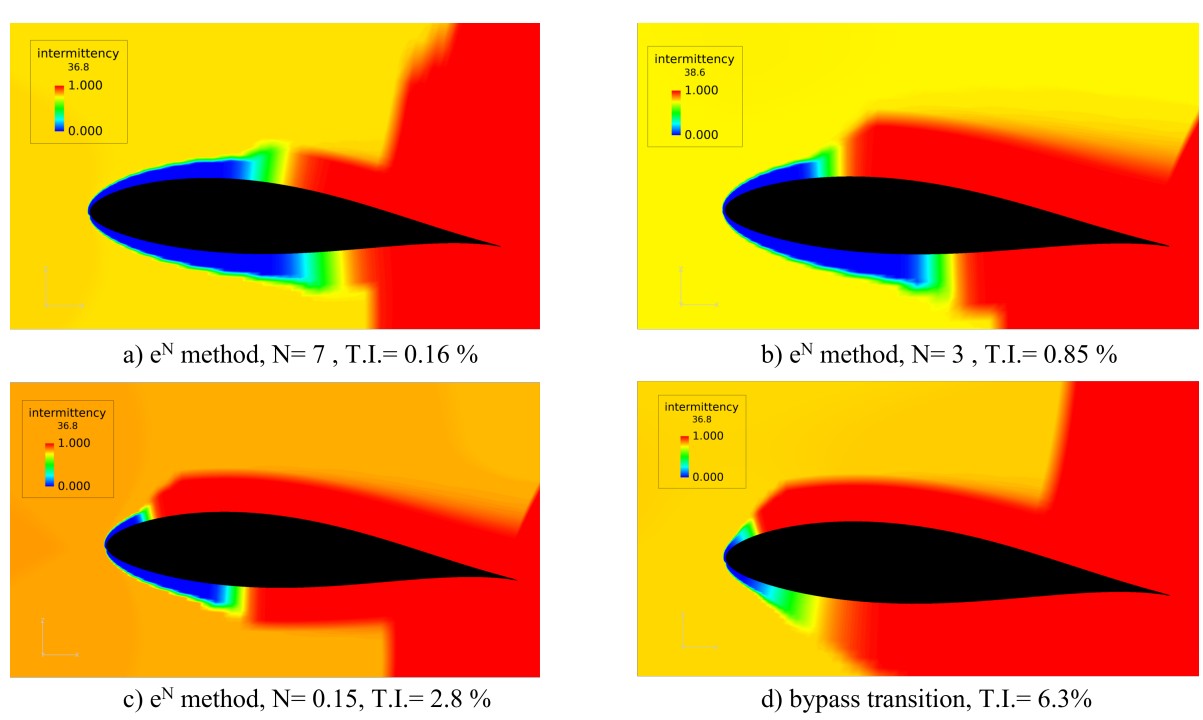

a) e$^N$ method, N= 7 , T.I.= 0.16 %

b) e$^N$ method, N= 3 , T.I.= 0.85 %

c) e$^N$ method, N= 0.15, T.I.= 2.8 %

d) bypass transition, T.I.= 6.3%

**Figure 12.** Intermittency contours for the case at $U_\infty$ = 6.3 m/s and $p = 4.75$ ° for the blade section at 36.8 m from the hub. The blade section results are obtained from the EllipSys3D URANS full rotor simulations.

A significant number of full rotor simulations were conducted with different transition models ($e^N$ and bypass transition)

and inputs for the free-stream velocity, pitch setting and inflow turbulence levels. In the post processing, data from the section



of one of the blades, which was instrumented with high-frequency microphones in the field experiments, is analyzed. The effective angle of attack in the simulations is calculated by the annular averaging of the axial velocity method in order to compare the results with the field experiments. The details of this method and how it is implemented can be found in previous studies (Hansen et al., 1997; Özçakmak, 2020).

An example case for a free-stream velocity of 6.3 m/s and $T.I. = 6.3\%$ inflow conditions, where the turbine operates under the effect of a wake of an upstream turbine (69% wake-affected rotor area) is shown in Figure 13. Several simulations were performed, since during a single revolution, the angle, inflow velocity and turbulence that the blades go through changes.

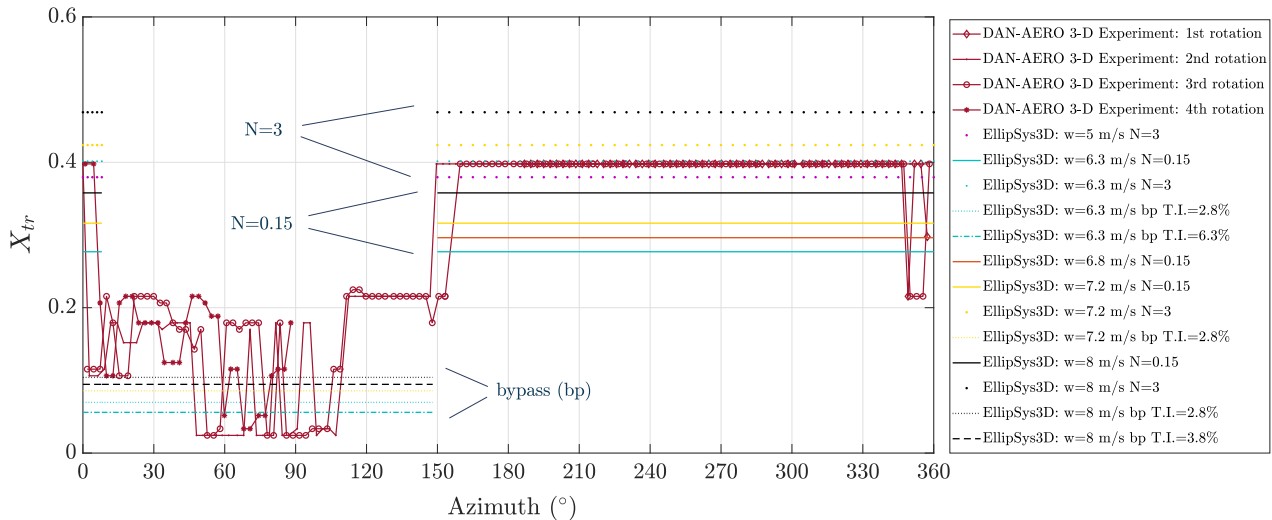

**Figure 13.** Experimentally detected transition locations as a function of the azimuthal angle for the case at $U_\infty = 6.3$ m/s, $T.I. = 6.3\%$ and $p = 4.75\,^\circ$ showing 4 revolutions. The EllipSys3D simulation results are labeled by the colors for various free-stream velocities, i.e., black: $w = 8$ m/s, yellow: $w = 7.2$ m/s, orange: $w = 6.8$ m/s, turquoise: $w = 6.3$ m/s, purple: $w = 5$ m/s. (N=0.15, N=3 and N=7 correspond to T.I. values of 2.8%, 0.85%, and 0.16% respectively.).

It is seen that at the azimuthal angles where the wake-affected inflow is present on the blade section, the experimental results agree with the simulation results obtained with the bypass transition model. Moreover, the natural transition zones
show agreement with the semi-empirical $e^N$ method with $N = 3$ for exactly the same free-stream velocity of 6.3 m/s as in the experiments. The regions where the transition point is detected around 20% of the chord are possibly due to the decreasing angle of attack on the pressure side. On the other hand, the regions where transition is close to the leading edge as also predicted by bypass transition URANS simulations, are the direct effect of the inflow turbulence in addition to the effect of the decreasing angle of attack.

Vorticity contours colored by the $w$-velocity component (free-stream velocity direction) obtained from the EllipSys3D full rotor simulations are shown in Figure 14-left for $w = 6.3$ m/s, $e^N$ transition method with $N = 3$ ($T.I. = 0.85\%$). Furthermore, the intermittency contours on the blade for a case at $w = 6.3$ m/s are presented in Figure 14-right. Each blade location represents




different inflow conditions. When the blade is at the top position (azimuth angle of $0°$), and when at $225°$, the blade is operating out of the wake. Therefore, transition line results obtained from $e^N$ method ($N = 0.15$ and $N = 3$, respectively) are shown for

those cases. Moreover, for the blade passing through $135°$ azimuthal position, the transition line calculated from the bypass transition model ($T.I. = 6.3$ %) is presented as it is operating under the wake of an upstream turbine at this position. All the results are gathered in a single rotor image and in this way, the changes in the transition line during a single revolution due to the changes in the angle of attack and the inflow turbulence are visualized (Figure 14-right).

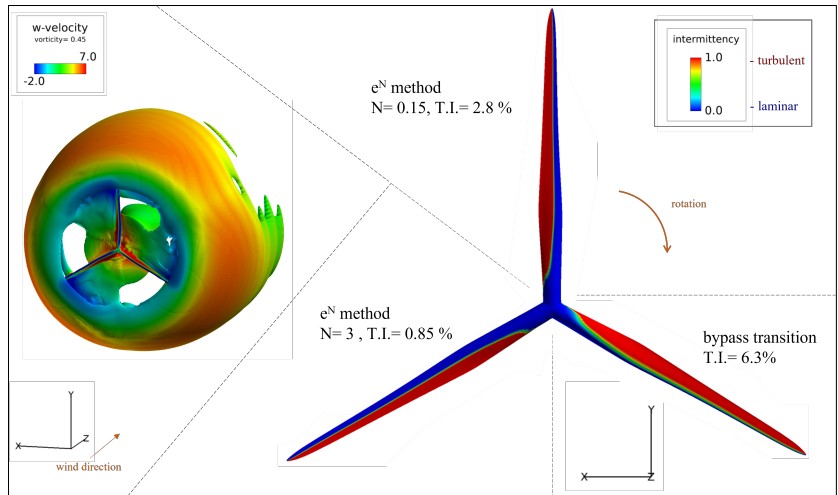

**Figure 14.** EllipSys3D-vorticity iso-surfaces colored by the $w$-velocity (wind direction) component (on the left) and intermittency $\gamma$ contours (on the right) obtained for three different inflow turbulence levels with natural and bypass transition models, shown in the same figure for representing the change of the transition line location throughout a single revolution ($w$ = 6.3 m/s).

Analysis of several measurements from different days with various inflow velocities and turbulence conditions as well as

two different pitch settings are collected in a single figure (Fig. 15) together with the EllipSys3D simulation results. Angle of attack values derived from the force measurements in the DAN-AERO field experiments and detected transition locations from the high-frequency microphones are presented in this figure, grouped according to the pitch angle and wake-affected inflow conditions. The EllipSys3D results with different transition models and inputs are also presented for both pressure (Fig. 15 - left) and suction sides (Fig. 15 - right). It is seen from the pressure side that within a single revolution, if the

inflow wake-affected rotor area is large, then the change in the transition location is scattered along $44\%$ of the chord on the pressure side. On the contrary, in the low angle of attack region, when the wake-affected rotor area is small, the variation of the transition location along the chord during a single revolution drops to $5\%$. The variation of the transition location during a single revolution is represented by separate EllipSys3D simulations conducted at various inflow conditions. EllipSys3D results for various amplification ratios can cover the range of the transition locations detected from the experiments. The experiments

show an earlier transition compared to simulations for natural transition on the pressure side at low angle of attack values. For higher angles of attack, on the other hand, the location of numerical transition fits to the results where there is natural





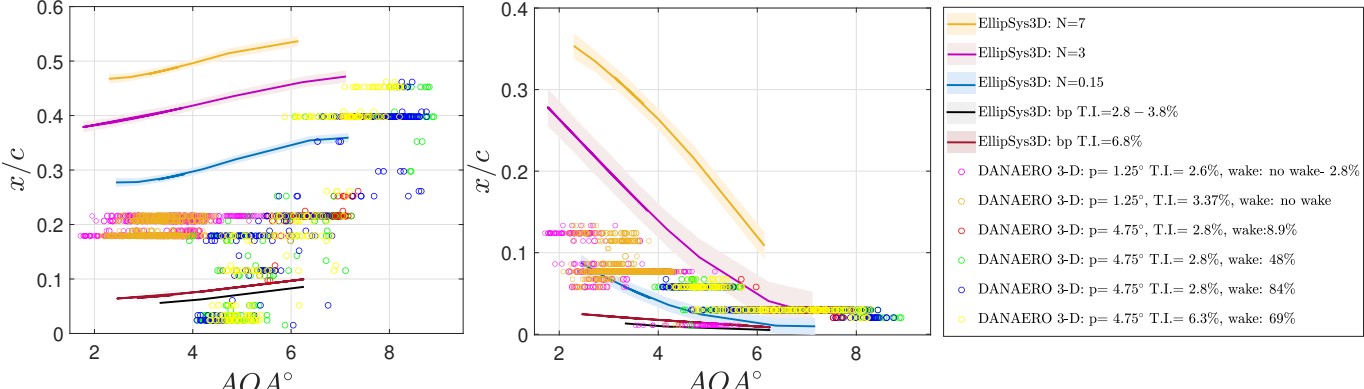

**Figure 15.** Detected transition locations from the EllipSys3D computations and the DAN-AERO field experiments for the pressure (left) and suction (right) sides, regrouped and regenerated from Özçakmak et al. (2020).

transition. For the bypass transition case, EllipSys3D results are within the range of the experiments. It is seen that both natural and bypass type transition mechanisms can occur in a single revolution of the turbine. Looking at the experimental results for a pitch angle of $4.75°$, for the same turbulence intensity value of $2.8\%$, as the wake-affected rotor area increases from 9 to 48

% (red dots to green dots), it is seen that the transition position starts to scatter more around the chord and bypass transition cases are observed for $x/c$ positions very close to the leading edge. Another comparison can be done between a low $T.I.$ of $2.8\%$ value and a high wake case (blue dots) and high $T.I.$ case of $6.8\%$ with similar wake conditions. A shift in the transition graph is observable through lower angles of attack, meaning earlier transition is observed at a certain angle of attack moving from blue to yellow dots, i.e., from the low to the high $T.I.$ case.

It is relatively harder to comment on the suction side as the detected transition positions from the experiments are in very close proximity to the leading edge (varying from $1\%$ to $13\%$). Moreover, some surface bumps that are visible from the pressure tap measurements could have played a role in changing the transition location on the suction side. However, it is still possible to see a good agreement between the simulations and the experiments.

### 3.3 Large-eddy simulations for the Aerodynamic Glove experiment

As a follow-up to the Aerodynamic Glove experiment on the Senvion (REpower) MM92 wind turbine (Reichstein et al., 2019), wall-resolved large-eddy simulations (LES) are carried out to provide more insight into the transition phenomena. The profile considered corresponds to the section of the LM 43P blade at a radius of 35 m and a relative thickness of 20 %. The simulations focus on a fixed angle of attack of $\alpha = 4°$ which lies within the range measured during the experiment. The Reynolds number based on the chord length $c$ and the free-stream velocity $U_\infty$ is $Re = 10^6$. A parameter study with varying inflow turbulence

intensities ($T.I.$) is carried out as indicated in Table 4. All units with respect to the LES predictions are non-dimensionalized using the chord length and the inflow velocity.





For the LES the filtered three-dimensional, time-dependent Navier-Stokes equations for an incompressible fluid are solved based on a finite-volume method on block-structured grids, which is second-order accurate in space and time (Breuer, 1998, 2000, 2002). A blended scheme is used in space which is a combination of a standard central difference scheme (95 %)and a standard upwind

scheme (5 %), whereas the time marching within the predictor-corrector method uses a low-storage Runge-Kutta scheme. In the present study, the widely used dynamic version of the classical Smagorinsky (1963) model is applied which was introduced by Germano et al. (1991) and Lilly (1992). The subgrid-scale stress tensor mimics the influence of the non-resolved small-scale structures on the resolved large eddies.

A C-type grid with the angle of attack already included in the base mesh is used. The grid extends eight chord lengths

upstream of the airfoil and fifteen chord lengths downstream of the trailing-edge avoiding the influence of the outflow boundary condition on the flow around the airfoil. This is a standard domain size which is sufficient for LES as also seen in, for example Gao et al. (2019) and Solís-Gallego et al. (2020). A suitable choice for the spanwise extension is critical for such geometrically two-dimensional airfoils. A width of $z/c = 0.06$ is chosen for the present Reynolds number, which is sufficient as discussed in Lobo et al. (2021). The grid has a wall-normal distance of the first cell center from the wall of $y_{1st}^+ < 0.5$. The streamwise

resolution is $\Delta x^+ \leq 30$ on the suction side, $\Delta x^+ \leq 60$ on the pressure side and $\Delta z^+ \leq 25$ for the spanwise direction. This satisfies the requirements for a wall-resolved LES as outlined by Piomelli and Chasnov (1996) with a $y_{1st}^+ < 2$ for the wall-normal resolution, $\Delta x^+ = \mathcal{O}(50-150)$ for the streamwise resolution and $\Delta z^+ = \mathcal{O}(15-40)$ for the spanwise resolution. The same applies for the conditions proposed by Asada and Kawai (2018), who performed a grid-convergence study and found grid-independent results for $c_p$ and $c_f$ as well as a sufficient resolution for resolving streaks using a grid with $\Delta x^+ = \mathcal{O}(25-50)$

for the streamwise, $\Delta z^+ = \mathcal{O}(13-30)$ for the spanwise and $y_{1st}^+ = 0.8$ in the wall-normal direction, respectively.

The generated inflow turbulence is anisotropic in nature and follows the Kaimal spectrum (Kaimal, 1973) as seen in Eq. (4), where $\sigma$ is the standard deviation, $L$ is an integral length scale and $\bar{U}$ the mean velocity at hub height. The input parameters used for the generation of the inflow turbulence are based on those suggested by the IEC-61400-1 standard and are shown in Table 5. According to this IEC standard, for hub heights greater than 60 m, the relative length scale $\Lambda_1$ = 42 m which

according to Table 5 leads to length scales of 340.2 m, 113.4 m and 27.72 m in the longitudinal, lateral and vertical directions, respectively. However, these scales are far too large for a computationally expensive LES and therefore they need to be scaled down to maintain the anisotropic nature such that the eddies have a similar form but not the same absolute size. Since the spanwise extension of the domain is the limiting dimension (0.06 chord lengths) the length scales are chosen based on this parameter. The maximum energy is located in the spanwise wavenumber $k_z = \sqrt{\pi}/L_z$ and the maximum spanwise wavelength

that can be resolved is based on the spanwise dimension such that $\lambda_z = 2\pi/k_z$ = 0.06. By solving these equations and using the relations from Table 5, it is found that the length scales in the three directions are 0.211, 0.07 and 0.017 dimensionless units, respectively.

$$E(f) = \sigma^2 \frac{4L/\bar{U}}{(1 + 6fL/\bar{U})^{\frac{5}{3}}} \ . \tag{4}$$

The inflow turbulence for the simulations is generated by a synthetic turbulence inflow generator based on the digital filter

method proposed by Klein et al. (2003) and improved for computational efficiency by Kempf et al. (2012). The necessary





**Table 4.** Flow configurations and observed transition locations using wall-resolved LES.

| Inflow Turbulence Intensity | Transition start | Transition end | Lift coefficient | Drag coefficient | Lift to Drag Ratio |
|---|---|---|---|---|---|
| 0 % | 52 % | 59 % | 0.4468 | 0.0076 | 58.79 |
| 0.6 % | 43 % | 53 % | 0.4400 | 0.0072 | 61.11 |
| 4.5 % | 3 % | 53 % | 0.4322 | 0.0076 | 56.87 |

**Table 5.** Kaimal length scales and standard deviation ratios from the IEC61400-1 IEC (2006).

| | Velocity component | | |
|---|---|---|---|
| | Longitudinal | Lateral | Vertical |
| Standard deviation $\sigma$ | $\sigma_1$ | $0.8\ \sigma_1$ | $0.5\ \sigma_1$ |
| Integral scale, $L_k$ | $8.1\ \Lambda_1$ | $2.7\ \Lambda_1$ | $0.66\ \Lambda_1$ |

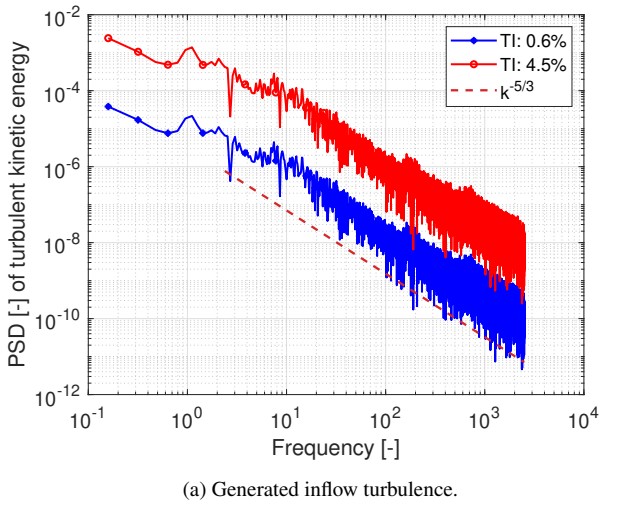

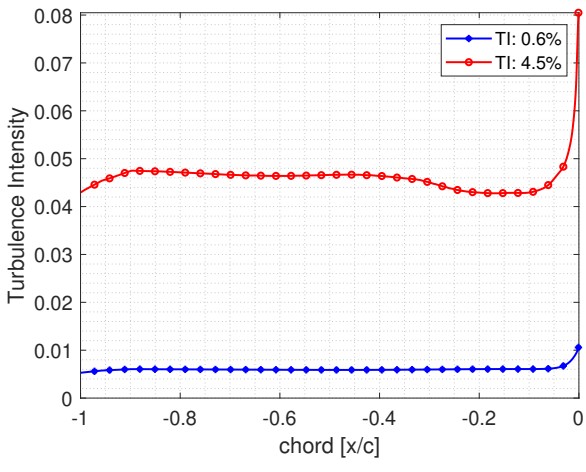

(a) Generated inflow turbulence.

(b) Turbulence decay within the domain.

**Figure 16.** Generated inflow turbulence and its decay within the domain along a line corresponding to the $x$-axis.

inputs are the Reynolds stresses, two integral turbulence length scales and one integral turbulence time scale. The inflow data is generated by multiplying filter coefficients based on the length scales with a series of random numbers. The necessary 3-D correlation between the filter coefficients is achieved by the convolution of three 1-D filter coefficients. Furthermore, the cross-correlations are taken into account guaranteeing the representation of a realistic inflow data satisfying the necessary one- and two-point statistics. The presently applied STIG only allows the definition of one length scale per direction. However, this disadvantage is compensated by superimposing the solutions of different length scales given by the maximal length scale divided by the factor $2^n$ ($n$ = 1-10) The generated turbulence (see Fig. 16a) is then inserted by a source-term formulation (Schmidt and Breuer, 2017) into the domain close to the region of interest to prevent numerical damping due to the coarse grid





resolution far away from the airfoil. This implementation has been thoroughly discussed in previous publications (Schmidt and Breuer, 2017; Breuer, 2018; De Nayer et al., 2018). Here the inflow turbulence is inserted one chord length upstream of the airfoil. The resulting turbulence decay is minimal as can be seen in Fig. 16b. The decay of the free-stream turbulence defined as $T.I. = \sqrt{\frac{1}{3} \times (\overline{u'u'} + \overline{v'v'} + \overline{w'w'})}/U$ is plotted using the averaged normal Reynolds stresses in the three principle directions (i.e., $\sigma_1^2 = \overline{u'u'}$ etc.) at the end of the simulation period. Here $U$ denotes the mean inflow velocity. The slight rise seen at about $x/c$ = -1 close to the injection plane is due to the insertion of turbulence based on a Gaussian bell-shaped distribution around the inflow plane. The required inflow turbulence is achieved slightly downstream of this plane. The rise near the nose of the airfoil located at $x/c$ = 0 is due to the influence of the airfoil. For the generation of inflow turbulence using the digital filter method mentioned above, the Reynolds stress tensor must also be determined in addition to the length scales. For a known $T.I.$, the turbulent kinetic energy (TKE) for an isotropic case can be determined by $k = \frac{3}{2}(U \times T.I.)^2$. Thus, knowing $k$ and using the relation $k = \frac{1}{2}(\sigma_1^2 + \sigma_2^2 + \sigma_3^2)$ together with the relations between $\sigma$ (Table 5), the Reynolds stress tensor for the anisotropic case can be determined. In order to allow for a $u$-$w$ correlation, a non-zero Reynolds stress term is included for this component. As shown in Jonkman (2009) this can be set as $-U_{star}^2$, where $U_{star}$ is the friction or shear velocity and has a typical value between 0.05 and 0.1 depending on the ground roughness scale. For the present simulations a value of 0.05 is arbitrarily chosen.

The results presented in this sub-section include selected aerodynamic properties, namely the pressure and friction coefficients, the shape factor and the Reynolds shear stress in the vicinity of the wall (Fig. 17), which will together help to identify the flow regime as laminar, transitional or turbulent. The graphs in Fig. 17 are plotted using the averaged flow over $8$ dimensionless time units corresponding to $2.68 \times 10^6$ time-steps with a time-step size of $3 \times 10^{-6}$. The data is additionally averaged in the spanwise direction. This is followed by snapshots of the instantaneous streamwise velocity disturbance for the visualization of boundary layer streaks in Fig. 18, plots of the Reynolds shear stress $\overline{u'v'}$ in the wall-normal direction to help identify the transition to fully turbulent flow in Fig. 19 and finally distributions of the power spectral density (PSD) of the turbulent kinetic energy (TKE) in Fig. 20 for a further investigation on the transition scenarios. Each inflow turbulence case will be separately analyzed, but comparisons will be made where necessary.

At an inflow turbulence of 0 % an obvious plateauing of the pressure coefficient is seen in Fig. 17a indicating the presence of a laminar separation bubble between 50 and 55 % chord. This is more apparent by the negative $c_f$ values in Fig. 17b. Here, transition to turbulence is clearly taking place between 52 and 59 % chord as indicated by the sharp drop and then rise in $c_f$ before its eventual reduction indicating fully turbulent flow. The shape factor which is the ratio of the displacement thickness to the momentum thickness as seen in Fig. 17c is also helpful in distinguishing between laminar and turbulent flow. The laminar, transitional and turbulent regimes for the $T.I. = 0$ % case are quite evident with the drop in $H_{12}$ between 52 and 60 % chord indicating transitional flow. This drop is due to the increase of the momentum thickness caused by shear indicating an increased momentum exchange which takes place during transition. This increase in momentum exchange is clearly seen in Fig. 17d, where the Reynolds shear stress $\overline{u'v'}$ at the second cell in the wall-normal direction on the suction side is depicted. At an inflow $T.I.$ of 0 % the Reynolds shear stress is zero along the suction side until a sharp rise of the magnitude is seen beginning at 52 % chord. A peak is found at 55 % chord and finally the magnitude reduces to a lower slope at approximately 60 % chord.





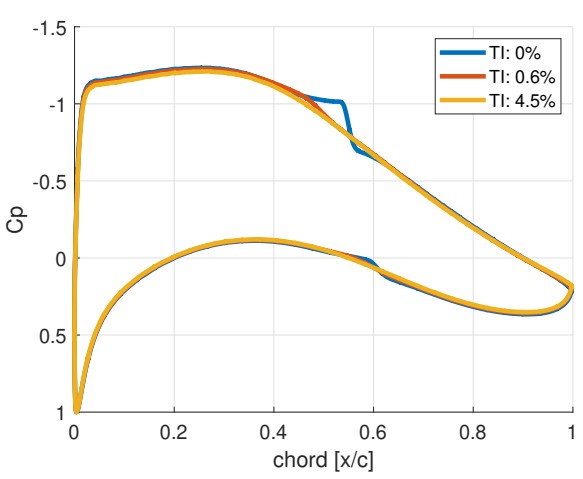

(a) Pressure coefficient based on the mean flow.

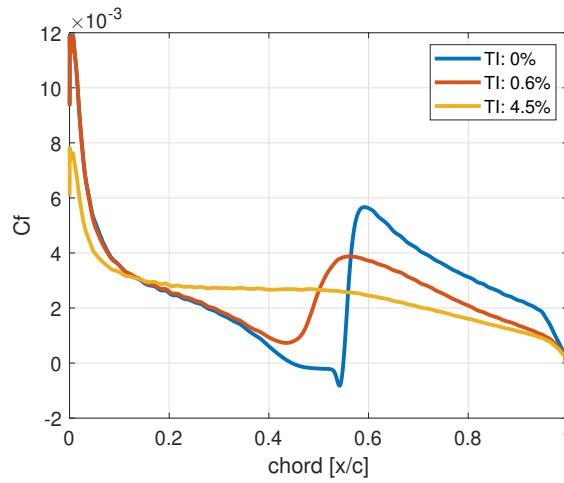

(b) Friction coefficient (suction side) based on the mean flow.

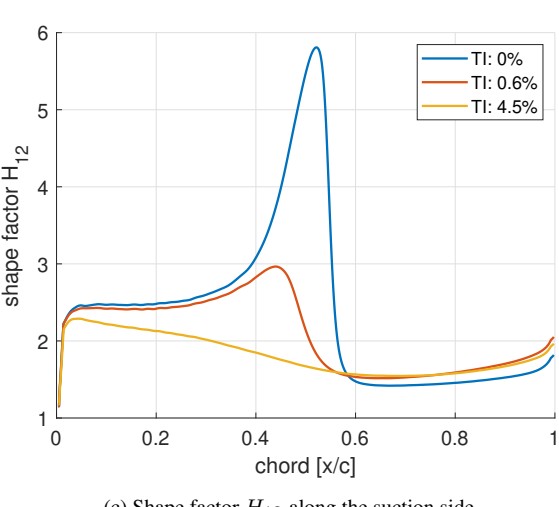

(c) Shape factor $H_{12}$ along the suction side.

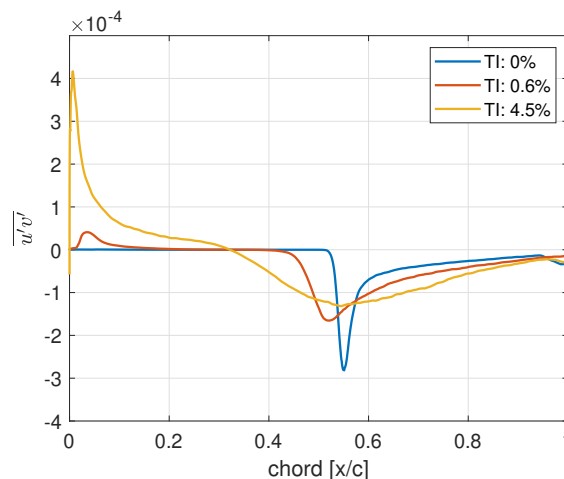

(d) Reynolds shear stress $\overline{u'v'}$ at the second cell normal to the airfoil wall on the suction side.

**Figure 17.** Distribution of the aerodynamic properties of the airfoil at a Reynolds number of $10^6$ and $\alpha = 4°$.

This transitional region between 52 and 60 % chord is in good agreement with that indicated by the shape factor. The peak at
55% chord corresponds in this case to the reattachment point but is also a good indication of what will further be referred to
as the peak transition to turbulence location. This can be visualized in Fig. 18a, where the instantaneous streamwise velocity
disturbance $u'$ at a height corresponding to the displacement thickness at 0.5 % chord is shown. Here at 56 % chord it is clearly
visible that the flow has turned turbulent. Upstream of this position at 52 % chord, the location indicated above as the beginning
of the transitional region also clearly shows a spanwise roll of the Kelvin-Helmholtz type further showing that it is possible to
distinguish the laminar and turbulent regions from the plots shown in Fig. 17.





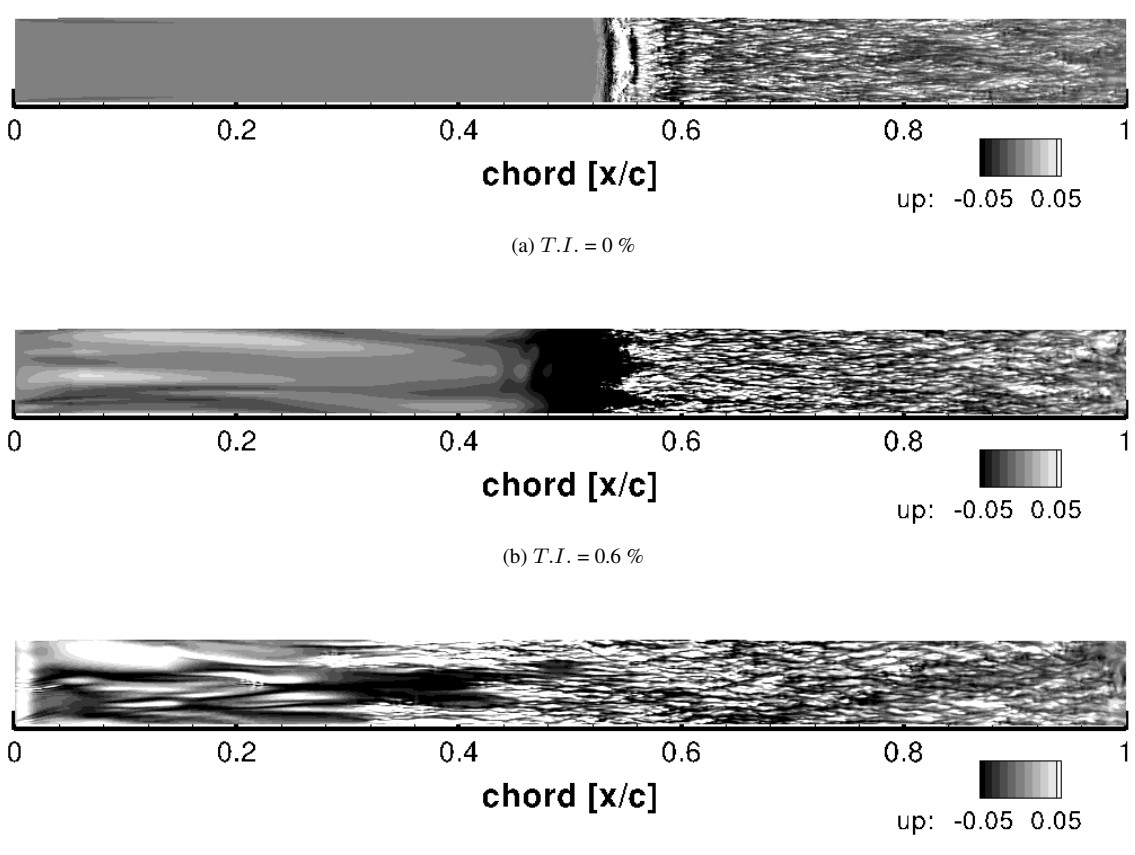

**Figure 18.** Snapshots of the instantaneous streamwise velocity disturbance $u'$ for the visualization of boundary layer streaks. Slices are taken at a wall-normal height corresponding to the displacement thickness at 0.5 % chord.

With an increase in inflow turbulence to 0.6 % there is no more an indication of separation in the mean flow, neither in the $c_p$ nor in the $c_f$ plot. An analysis of the instantaneous flow, however, does indicate the presence of a temporarily present separation bubble, which can be seen in Fig. 18b between approximately 48 and 52 % chord. From the plot of the friction coefficient (Fig. 17b) a clear distinction between laminar, transitional and turbulent flow regime is still evident with a laminar

region up to approximately 43 % chord. Downstream of this position the flow is transitional between 43 and 56 % chord indicated by the rise in $c_f$. Finally, the flow is found to be fully turbulent at 56 % chord. Transition to turbulence therefore moves upstream in the presence of added inflow turbulence. From the plot of the shape factor (Fig. 17c) a clear transitional region is once again visible and it is interesting to note that the flow has a similar value of the shape factor when fully turbulent at around 58 % chord onward as that seen at 60 % chord with $T.I. = 0$ %.

For the cases with added inflow turbulence, the plot of the Reynolds shear stress (Fig. 17d) is a good indicator of the receptivity of the boundary layer to the external disturbances, again due to the fact that this factor indicates momentum exchange.





At $T.I. = 0.6$ % a small peak is seen at 4 % chord extending up to about 10 % chord with a small but noticeable increase even up to 20 % chord. This case shows the shear-sheltering phenomenon (Hunt and Carruthers, 1990) wherein lower frequencies have the capability of penetrating deeper into the boundary layer compared to higher frequencies. Moreover, the leading edge having the thinnest boundary layer allows for penetration of a higher range of inflow disturbances (Jacobs and Durbin, 1998). This is a fact clearly indicated by the rise of the Reynolds shear stress near the leading edge with a reduction as the flow moves downstream showing a reduced interaction between the external flow and the boundary layer. Boundary layer streaks or Klebanoff modes are produced as a response of the boundary layer on account of the penetration of external disturbances. It is remarkable that even a low $T.I.$ of 0.6 % results in the formation of boundary layer streaks as seen in Fig. 18b. They are visible as elongated dark (slower than the mean flow) and light (faster than the mean flow) longitudinal regions of the flow. As discussed in Lobo et al. (2021) positive streaks tend to delay separation (seen around 40 % chord and at the bottom of the image) while negative streaks move separation upstream (seen above the positive streak at 40 % chord). Just as in the case without added inflow turbulence, for the case with a $T.I. = 0.6$ % the largest peak in the magnitude of the Reynolds shear stress at 53 % chord is the location at which the flow turns turbulent. This can further be confirmed by the plots of $\overline{u'v'}$ as seen in Fig. 19b where the profile begins to show the typical turbulent characteristics of a sharp peak near the wall followed by a region where there is an obvious change in the slope. The difference between 50 and 53 % chord is obvious. Similar plots are also seen at a $T.I. = 0$ % (Fig. 19a) where the double peak typical for the turbulent region is first seen at 59 % chord. This matches the expected chord location from the friction coefficient and shape factor estimates.

On further increasing the inflow turbulence to 4.5 % the $c_p$ curve (Fig. 17a) between 40 and 50 % chord somewhat flattens in comparison to the case with an inflow $T.I.$ of 0.6 %, i.e., the slight bump around the region where intermittent separation is observed is not present anymore. In this case no evidence of instantaneous separation is seen and this is probably on account of the increased frequency of streak formation within the boundary layer that prevents the formation of a separation bubble as seen in Fig. 18c. Here, turbulent bursts of the varicose kind (Vaughan and Zaki, 2011) are also seen. Overall, especially on the suction side in the laminar region, up to approximately 50 % chord a further decrease in the suction-side pressure coefficient is seen which results in a reduction of the overall lift with increasing inflow turbulence. Distinguishing between laminar, transitional and turbulent flow in this case is not as easy as compared to the cases with lower $T.I.$ discussed above. An increase in $c_f$ indicating transitional flow is not apparent in Fig. 17b as in the previous two cases. However, a relatively sharp drop is seen at approximately 50 % chord indicating a change in the flow characteristics, probably a sign of turbulent flow. Using the plot of the shape factor (Fig. 17c) as an indication, the extent of the turbulent regime is quite similar to that at $T.I. = 0.6$ % with the flow having similar characteristics beyond 56 % chord. There is once again no evidence suggesting a clear separation between the laminar and transitional zone. The flow seems to be already transitional near the leading edge at approximately 3% chord. On investigating the Reynolds shear stress at different chord locations depicted in Fig. 19c, a turbulent profile with a peak close to the wall and an obvious change in slope as discussed in the previous two cases is first seen at 53 % chord. This coincides quite well with the location of the peak in $\overline{u'v'}$ seen in Fig. 17d. Thus, once again this peak can be used to detect the point at which the flow turns turbulent. Furthermore, a similar trend in $\overline{u'v'}$ is seen where the greatest exchange of momentum is towards the leading edge and the penetration of disturbances as indicated by the exchange of momentum decreases along the





(a) $T.I. = 0$ %.

(b) $T.I. = 0.6$ %.

(c) $T.I. = 4.5$ %.

**Figure 19.** Distribution of $\overline{u'v'}$ in the wall-normal direction at selected chord locations.

chord. In this case positive values of $\overline{u'v'}$ are seen up to 32 % chord which is the region where the adverse pressure gradient begins and this also coincides with the highest region of the airfoil at this angle of attack. The boundary layer downstream of this location is somewhat shielded by the upstream presence of the airfoil, so it is not clear whether the adverse pressure

gradient is the reason for the decrease of this quantity to zero or if it is on account of slight sheltering by the presence of the airfoil upstream of this point. The negative quantity beyond this location indicates that there is once again momentum exchange between the boundary layer and the external flow with a mean exchange in the outward direction.





(a) PSD at $T.I. = 0$ %.

(b) PSD at $T.I. = 0.6$ %.

(c) PSD at $T.I. = 4.5$ %.

(d) Data output locations at the height of displacement thickness (shown here: case without added inflow turbulence).

**Figure 20.** Power spectral density of the turbulent kinetic energy at different $T.I.$ determined at the height of the displacement thickness in the mid-span at every 10 % chord.

For a further investigation on the mode and process of transition and the influence of the inflow turbulence on the boundary layer receptivity, it is helpful to analyze the power spectral densities (PSD) along the chord for the different cases under consideration. Figure 20 shows the PSD plots of the turbulent kinetic energy computed using a Hann windowing function on the data collected at a height corresponding to the boundary layer displacement thickness and at every 10 % chord. Data was collected at a dimensionless frequency of 8.3k covering 6 dimensionless time units. Figure 20d indicates the regions where these quantities are evaluated.





At an inflow turbulence intensity of 0 % a clear increase in energy is seen at 40 % chord with the separation bubble showing
a sharp increase in energy at a frequency corresponding to 10 dimensionless units at 50 % chord. The highest energy in the
spectra is then found at 60 % chord with a decrease as one moves downstream. The region where the flow first turns fully
turbulent is expected to have the highest turbulent kinetic energy (Özçakmak et al., 2019). At a T.I. of 0.6 % it is quite evident
that the energy at 10 % chord is approximately a factor of five higher than in the absence of inflow turbulence. At 40 % chord
there is a large increase in energy indicating that the flow is transitional and at 50 % chord the flow is turning turbulent as
discussed earlier. Once again, this is the chord location with the highest energy with a reduction seen as the flow proceeds
downstream. It must be pointed out that there is a small peak at 10 dimensionless units which corresponds to the peak seen at
$T.I. = 0$ %. The PSD being plotted from the time-series data also includes time steps wherein there is instantaneous separation.
This is the reason that a peak is seen even though the mean flow has no indication of separation.

On further increasing the inflow turbulence to 4.5 % it is nearly impossible to distinguish laminar and turbulent regimes.
However, at 10 % chord the energy content is obviously lower than at the other chord locations, but evidently higher than in
the previous cases. This is due to the fact that the higher frequencies also have higher energy compared to the previous case,
thereby allowing higher frequencies to penetrate deeper into the boundary layer. Furthermore, 40 and 50 % chord have a similar
and the highest energy indicating that this is the location around which the flow turns fully turbulent. This is in agreement with
the discussion above.



## 4 Summary on experimental findings and simulating transition on wind turbine blades

To optimize the aerodynamic design of wind turbine blades, it is essential that the transition process under varying conditions in the free-atmosphere is better understood, thereby paving the way for the creation or improvement of existing transition models. Precise prediction of the transition location will lead to the design of more efficient blades and avoid the present design procedure where an empirical based blend of airfoil data based on full turbulent and free transition simulations is used. For wind turbines in atmospheric flow and in the presence of inflow turbulence, especially in the wake of an upstream wind turbine, bypass transition is found to play an important role. The present study analyzes transition on the LM-38.8 blade of the 2MW NM-80 rotor from the DAN-AERO experiment (Madsen et al., 2010b) and the LM 43P blade of the Senvion (formerly RE-power) MM92 wind turbine corresponding to the Aerodynamic Glove experiment (Reichstein et al., 2019). Corresponding to these experiments unsteady Reynolds-averaged Navier-Stokes (URANS) simulations and wall-resolved large-eddy simulations (LES) are respectively carried out. Furthermore, an overview on the present capabilities of Reynolds-averaged Navier-Stokes (RANS) methods which are computationally the cheapest are presented with respect to wind turbine transition prediction thus showcasing the capabilities of computational fluid dynamics (RANS, URANS and LES) at different levels of fidelity.

### 4.1 Experiments

– From the experiments it is clear that even on turbines operating in the free atmosphere, laminar regions can be present and this can be detected from pressure taps, microphone arrays and thermographic imaging techniques.

– In case of bypass transition, inflow turbulence is found to be a predominant factor together with changes in the angle of attack during the revolution. The role of surface roughness cannot be ruled out or quantified.

– The DAN-AERO experiment shows that the transition scenario changes even over the course of a single revolution due to the changing characteristics of the upstream flow field (change in angle of attack, shear, and upstream wake conditions). In this particular study, natural transition was found to dominate when the turbine operated outside the influence of an upstream wake, whereas in the presence of a wake, bypass transition was observed. Furthermore, boundary layer recovery is evidently quite quick with natural transition taking place at azimuthal angles directly outside the influence of the wake. The transition locations are detected by integrating the spectra in the range of 2-7 kHz and determining the highest derivative of this integration above a threshold. It is also observed that the type of the transition can be estimated by the power spectral behavior from the amplification of the spectral energy around particular frequencies.

– A high-frequency microphone placed at the laminar boundary layer region close to the leading edge and integrated over a low frequency range (100-300 Hz) provides information about the inflow turbulence level and this can be linked to the transition behavior. In the case of natural transition three stages, namely, receptivity, linear amplification and non-linear interactions (secondary-instabilities) could be visualized through power spectral density plots of the fluctuating pressure field.





- With respect to the Aerodynamic Glove experiment, short experiment times and a non-perfect surface has led to only few data sets with large areas of laminar region. However, it can be concluded that at least the same accuracy of microphones and the two thermographic techniques with respect to the transition location is found.

## 4.2 3D-RANS with transition modeling

– From this simplest and computationally cheapest approach it can be seen that with a reasonable choice of the amplification factor ($N$) corresponding to the the well known $e^N$ model for the growth of disturbances, a transition line (see Fig. 10) indicating a laminar flow region can be predicted for steady flow neglecting wind shear and other vertical changes of the inflow conditions.

- Deviations between the transition locations were found for the different codes tested with transition being detected
between 20 and 40 % chord for the test case discussed. Transitional models showed larger deviations in the power coefficient $c_P = 0.482 \pm 0.015$ when compared to the fully turbulent models which showed a much smaller deviation of $c_P = 0.445 \pm 0.005$.

## 4.3 URANS

- On separately calculating and analyzing various configurations of URANS simulations by using DTUs in-house CFD
solver EllipSys3D which simultaneously checks for natural and bypass transition, it is shown that there is good agreement between simulations and experiment with natural transition in the absence of an upstream wake and bypass transition in the presence of a wake.

- High turbulence levels seem to trigger bypass transition in the boundary layer in addition to its effect on the effective angle of attack. The chordwise location of bypass transition detected from the simulations is within the range of the
experiment.

- With respect to natural transition and on the pressure side, earlier transition was observed in the experiments when compared to the simulations at lower angles of attack, whereas at higher angles of attack the location of the numerical transition between the simulations and experiment tend to agree. It is relatively harder to make similar arguments on the suction side as the detected transition location from the experiments are in close proximity to the leading edge (between
1 and 13 % chord). Surface bumps visible from the pressure tap measurements could have also played a role in transition. However, it is still possible to see a good agreement between experiments and simulations.

- Similar to the 3D-RANS simulations, a justified choice of the amplification factor ($N$) is difficult to choose without experimental data to compare with. In this case an $N$ factor between 0.15 and 3 provides reasonable results.





## 4.4 LES

— The wall-resolved LES is successful in resolving various transition scenarios without any additional input such as an amplification factor $N$, nor does it depend on an intermittency factor as in the case of bypass transition using URANS methods.

— At low inflow turbulence intensities it is quite easy to distinguish between laminar, transitional and turbulent regimes from either the friction coefficient or the shape factor. With increased inflow turbulence, however, the flow seems to be 525 transitional from close to the leading edge and fixing a point where the flow turns turbulent is not as easy. Nevertheless, this is possible from the Reynolds shear stress distribution $\overline{u'v'}$ close to the wall with the flow turning fully turbulent at the most negative values of $\overline{u'v'}$ due to the highest exchange of momentum at this point on account of transition.

— The shear-sheltering effect is also evident based on the Reynolds shear stress distribution with disturbances penetrating into the boundary layer over a larger chord length when the inflow turbulence is increased due to the augmented energy 530 of the flow.

— The power spectral density plots of the turbulent kinetic energy at the height of the displacement thickness are quite useful in determining the transition scenario and are comparable to those from the DAN-AERO experiment with a clear growth of disturbances in the case of natural transition and higher energy in the low-frequency spectra in case of bypass transition. The PSD plots can also be used to determine the location of transition provided that a sufficient number of data 535 points are recorded along the chord. The highest energy on the PSD of the TKE coincides with the transition location. The same is also seen from the wind tunnel study on the model blade section from the DAN-AERO experiment. This once again can be attributed to the highest exchange of momentum and thus fluctuations within this region.

— The increase in energy in the lower frequency range with increasing inflow turbulence is comprehensible. The penetration of these low frequency disturbances gives rise to boundary layer streaks which can also be visualized. With increasing 540 inflow turbulence the number of streaks and their intensity increases. The spanwise wavelength does not seem to be influenced by the turbulence intensity alone and most likely depends on the length scales of the external disturbances.

— In summary, LES enhanced by artificial anisotropic turbulent inflow mimicking flow conditions of the free atmosphere even at high Reynolds numbers $\geq 1$ M is possible and feasible with present-day high-performance computing technology.

Overall, from the experiments and simulations it is seen that the laminar to turbulent transition and the different scenarios 545 through which it takes place can be detected through experiments and predicted with reasonable accuracy through simulations with higher fidelity simulations being more reliable and providing data which can be used to better understand the underlying physics of the different transition scenarios. To advance further our insight into the fundamental transition mechanisms of wind turbine blades, more dedicated experiments measuring the transition process on blade sections in atmospheric flow are needed, e.g. quantifying the impact of the inflow turbulence intensity. In parallel high-fidelity LES simulations should also be advanced 550 to a level where the experimental cases can be directly simulated.





*Author contributions.* APS, BAL, OSO, NNS and HM participated in the IEA Task 29. HM was project coordinator of the DAN-AERO experiments, OSO conducted the data analysis for boundary layer transition and interpreted the DAN-AERO results combining it with theory. NNS provided ideas, concept and guidance for the URANS simulations, OSO performed the URANS CFD simulations and post-processed and presented the results. APS was involved in the Aerodynamic Glove experiment. BAL performed the LES simulations and

555 its analysis under the guidance of MB (providing additionally the LES and STIG codes) and APS. BAL, OSO and APS wrote the draft manuscript and MB, HM and NNS reviewed and edited the manuscript.

*Competing interests.* The authors declare that they do not have any conflicts of interest.

*Acknowledgements.* RANS and LES computations by members of KUAS were performed with resources provided by the North-German Supercomputing Alliance (HLRN). Special thanks from KUAS go to Galih Bangga (IAG, now: DNV, Bristol, UK) and Leo Hönig (FOR-

560 wind/IWES, Germany) for numerous discussions and trials on mesh conversion to be used in RANS simulations. The DTU contribution has partly been carried out within the Danish participation IEA Task 29 IV project based on funding from EUDP 2018-I, contract J.nr. 64018-0084. Other contributions have come from DTU through funding of a PhD student, that has participated in the IEA task 29 IV.



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
