# Peer review of "On the laminar-turbulent transition mechanism on Megawatt wind turbine blades operating in atmospheric flow"

_Wind Energy Science, 2022_

## Referee Comment (RC1)

**General**

- I note that the article is written by different authors with their own style and notations e.g. sometimes y as radial coordinate (line 233), sometimes z(figure 10). For the free stream velocity, I sometimes see v_in (line 200), sometimes it is U_infty (figure 12). In figure 13 you even use 2 two different notations for the free stream velocity in the same figure: U_infty and w.
  An overarching coordination effort to make the different chapters more consistent, not only editorial but also in terms of structure, level of detail and content would improve readability a lot. I see chapters on two different experiments and 3 different models with a nice summary in section 4 but also from this summary the chapters seem mostly independent to me and a reader will wonder how they are connected, i.e. what is the 'red line'. Maybe there are not so many connections (apart from the fact that it is all about boundary layer transition on wind turbines) but then that needs to be explained in the introduction.

- Another comment is the definition of turbulence intensity i.e. the standard deviation of velocity fluctuations normalized by mean velocity. On a rotating wind turbine blade, in particular at the outer part the normalization velocity is largely driven by the rotational speed. At the same time the spectrum of the turbulent velocities on a rotating blade section is different than the turbulent free stream spectrum. I donot see this aspect mentioned somewhere and between the lines it seems to me that you assume the turbulent intensity and spectrum on a blade section to be similar to the free stream turbulent intensity. Maybe it is not so relevant and it is just a matter of definition but I am not sure if this is true when you apply the Mack relation. I want to be sure that you donot apply the free stream turbulence intensity for a rotating blade section. Could you elaborate?

**Then there are several small comments:**

- Title: It says transition mechanisms on **Multi MW** wind turbine blades. The results are given for Multi MW turbines indeed but Multi MW can range from 2 to \infty where the results of your research seems limited to 2 MW's turbines (Actually I cannot find the rated power of the turbine on which the glove experiment is done but I guess it is around 2 MW).

- Table 1: Could you add the rated power in the table. (As mentioned above the title of the article refers to rated power, then I would expect that the rated powers of the turbineare mentioned in the text)

- Table 1: Could you tell something on the airfoils involved?

- Table 1: The following transition experiment is done in the field too so it may have to be mentioned? *Experimental investigation of Surface Roughness effects and Transition on Wind Turbine performance O Pires et al 2018 J. Phys.: Conf. Ser. **1037** 052018*

- Line 72: Could you elaborate a bit more on the measurements. In particular the number of microphones and pressure taps. I suggest to add a statement that the frequency response of the pressures from the taps is relatively slow.

- Figure 1 is very interesting but I found it difficult to understand. This is partly due to my color blindness (which you may ignore) but also due to the explanatory text. For example how

does a reader recognize laminar spectra with apparent small peaks. Which small peaks? Can you clarify, possibly by indicating these peaks in the figures itself. It also helps to indicate the the transitional spectra with T-S wave peaks in the figure.

- Figure 2 is difficult for me too. There are very many lines for different conditions but why do you need so many conditions, what is the message of this figure? You make use of dashed and solid lines but that distinction is not visible anymore at the high frequency. Would it be an idea to divide the results over two figures to improve readability.

- The legends of figure 6 say: x/c_tr. I would make it x_tr/c

- Section 2.2: You mention that the experiment was intended to be done with microphones but it was enhanced with ground based thermography. Was it enhanced or replaced? In other words do you have results from microphones and if so why aren't they presented?

- Section 2.2. The description of the measurement technique is very limited. Could you add a few words on what ground based thermography means. How does the distance from the ground to the blade plays a role. These are averaged measurements? How many revolutions? Also why do you need two teams. Did they use different techniques in what respect?

- Line 163 How do you know it is equal accuracy, I donot see scientific evidence for that? Is there some kind measurement accuracy analysis carried out? The technique seems to find a transition location indeed but in theory it could a wrong location.

- Line 163 This suggests as if this technique is of equal value than the microphone technique but that is not true: Information on the spectrum is lacking.

- Line 169. *Prdmdl* looks a bit weird for unfamiliar readers. From the text I understand it is a transition model but could you add a few explanatory words or a reference?

- Section 3.1: Overall I find this section a bit difficult to follow. Amongs others due to the following:

  - I donot like the title *Findings from IEA Wind TCP Task 29 Subsection 3.6* very much. Could you make it more descriptive e.g. by mentioning DanAero or something like that? Apart from that it is not 100% correct. It is not subsection 3.6 from the report. but Task 3.6, described in chapter 10. Moreover, although I am honored that you mention me to be the author I feel it is more fair to mention the authors of this particular chapter: A. P. Schaffarczyk, B. A. Lobo, H.A. Madsen, O. S. Ozcakmak

  - Table 2: Meshes of what? I assume it is for the DanAero blade but I donot think that is mentioned somewhere in the text.
  - Line 194: It may be a bit uncommon to mention individuals. You already acknowledge Leo at the end of the article so that could be enough.
  - Line 196: NN-2013 what is this?

- Line 193: On Line 185 you mention Ellipsys, Flower and Fluent but now OpenFoam is mentioned as well?
- Line 200: Case I of IEA Task (vin = 6.1 m/s and 12.3 rpm) is cryptic. You anyhow need to mention that it is for DanAero, I think you also need to mention it is for axi-symmetric conditions.
- Figure 10 shows a lot of results which come out of the blue to me and which I cannot relate to the previous text where you prepare me for results from Ellipsys, Flower and Fluent and OpenFoam.
- Line 203: What is Fluent *Bra* I have never heard of *Bra?*
- Line 205 and further seems to explain the results from table 3 but I find it very difficult to follow. You mention results from Siemens, LM and DTU but if I understand it correctly they come from separate study. What is the relation to the present study? You mention a CP of 0.482 +/- 0.015 with transition and 0.445 +/- 0.005 but table 3 seems to give different results.
- Figure 10: Transition instead of transition
- Figure 10: I donot think everybody understands the names mentioned in the legend? This needs further explanation.
- Table 3: I would not call wt_perf an outdated BEM code. BEM is BEM for such simple case.
- Line 217: You could add that is not only CP but also CT which is lower. An error in pitch larger that 0.4 degrees could explain both (Here I am just speculating).
- Line 227: How does that work? You prescribe a T.I. value as input to the bypass model but I assume that is std. deviation normalized to the local mean velocity, in other words I assume you donot use the free stream turbulence intensity?
- Line 233: I would mention the relative radial position instead of absolute y value
- Figure 13: Please indicate that is a blade section 36.8 meter from the hub like you do in figure 12.
- Line 257: Do you have some kind of estimate for changes in wind speed, turbulence intensity, and angle of attack which the blade section at y = 36.8 undergoes during a revolution. That would help interpretation even it is a rough indication only
- Figure 14: Maybe good to add that we look to the pressure side
- Line 271: How do you derive aoa from force measurements? That is far from trivial. Could you elaborate a bit.
- Section 3.3 You suggest as if the calculations are done on the Glove experiment but it seems to me a non-rotating blade section or am I mistaken?
- Line 298: I donot think that relative thicknesses of airfoil are mentioned in the other sections (I would suggest to do that)
- Page 19 and further: I think that the level of detail of this model description is higher than the level of detail for the model descriptions of the previous section. I realize this is partly due to the fact that you add inflow modelling but I still have the idea it is a bit out of line. Please check and if possible/needed try to improve it.
- Line 340: STIG what is that?
- Figure 17: Title says THE airfoil. There are many airfoils in this article so please indicate.
- Figure 18: Please put a label in the vertical direction (I assume that is spanwise direction?). The legend is unclear to me: *"up: …"* ??? I assume we see the suction side?
- Line 398: Lower frequencies of inflow turbulence?
- Line 403: Klebanoff modes: What is that? Can you refer?

- Line 475: I donot think I have seen evidence of laminar flow derived from measured pressure distributions?
- Line 500: On suction side?
- Line 502: But you have much less fully turbulent models isn't it? So that could lead to a smaller deviation?

Good luck!

Gerard Schepers

---

## Referee Comment (RC2)

**Summary:** 'On the laminar-turbulent transition mechanism on Multi-Megawatt wind turbine blades operating in atmospheric flow' examines the role of inflow conditions and turbine operation on the blade boundary layer transition from laminar to turbulent flow. The authors consider studies across the fidelity spectrum including wind tunnel experiments and field campaigns as well as RANS, URANS, and LES simulations. They identify the turbulent transition point in response to various conditions in each study and provide two main mechanisms for boundary layer transition.

**Key Points:** The authors summarize the key findings from multiple studies on blade boundary layer transition and discuss the impact of ambient turbulence intensity and turbine operation on promoting either natural or bypass transition across the blade. The authors use multiple quantities of interest to support their findings and provide direct comparison between experimental and simulated results where possible. Linking boundary layer behavior to inflow turbulence intensity is an important result for wind turbine operation and the design of wind plants. Additionally, the relationship between blade azimuth and transition point is an interesting finding. Overall this paper has potential to become an authoritative reference on blade boundary layer transition but is held back by inconsistent writing and presentation styles.

**Recommendation:** I recommend publication in *Wind Energy Science* provided the authors successfully address the suggestions detailed below.

**General Suggestions:**

1. I strongly recommend general language editing to achieve a consistent writing style. The content is sound but the tone and writing change abruptly throughout the manuscript. It would benefit the article to have a unified presentation.

2. Figure formatting varies dramatically between sections. I understand several figures are referenced from prior works but where possible, a cohesive presentation style would assist the reader.

3. Section transitions are abrupt, not only is the writing style different but the focus of discussion changes as well. Reported quantities also differ with each section. While each quantity is related to transition, the manuscript feels disjointed. Transitory paragraphs between the various sections and methodologies would be a welcome addition.

4. I appreciate the direct comparison between experimental observations and simulated flows in Figures 13 and 15.

5. The summary in Section 4 is excellent.

**Specific Suggestions:**

1. A major finding is how transition is affected by inflow turbulence. The authors observe different transition points in Section 2.1 and briefly mention this difference in Section 4. However, turbulence produced by an upstream turbine, as was the case in the DAN-AERO experiments, will have a different structure than that of an active grid or synthetic turbulence generated in simulation. Can the authors comment on how this might impact their findings, particularly in relation to the PSD presented in Sections 2.1 and 3.3?

2. Consider placing the diagram in Figure 4 ahead of Figure 2 since this result also considers different wake overlap scenarios.

3. Revise Figure 11. The legend is obscured and the title is unclear.

4. It is not immediately apparent why some quantities are presented i.e. vorticity in Section 3.2.

5. Are $x/c$ in Figure 5 and $X_{tr}$ in Figure 13 the same? Where is $X_{tr}$ defined?

---

## Author Comment (AC1)

**On the laminar-turbulent transition mechanism on Multi-Megawatt wind turbine blades operating in atmospheric flow**

B.A. Lobo, O.S. Özçakmak, H.A. Madsen, A.P. Schaffarczyk, M. Breuer, N.N. Sørensen

**Review # 1**

We appreciate the effort of Gerard Schepers for evaluating our manuscript in detail. In the following his remarks are answered and modifications resulting from his comments are explained. Note that in the annotated version of the manuscript all modifications (replacements, additions and deletions) regarding the remarks of reviewer # 1 are highlighted in red.

Please note that in response to both reviewers regarding a consistent writing style, the text was in general revised to achieve this by a single author. No highlights are made to such changes as the message delivered and information contained remains unaltered.

**Response to specific comments:**

- **Free-Stream Velocity definition throughout the text**
  We have changed for the consistency of the definition of the free-stream velocity to 'w' in the paper. That is not highlighted by red in the text. In the case of the LES simulations, however, it was necessary to retain the coordinate system used, but in this case the notations are made clear to prevent any confusion.

- **What is the red line**
  As you guessed, the red line is mainly that all the studies deal with boundary layer transition on wind turbines with some comparisons that have already been made when applicable. The focus is on the impact of the atmospheric turbulence, which makes the conditions quite different from wind tunnel investigations. The paper synthesizes information on wind turbine rotor transition from two main full scale experiments and supported by numerical simulations of rotor transition with CFD codes of different fidelity, all indicating that a mix of natural transition and bypass transition is present. This has been clarified in the last paragraph of the introduction just before discussing what to expect in the different sections of the paper.

- **Definition of turbulence intensity**
  In case of the URANS simulations of the DAN-AERO blade, the turbulence intensity is calculated from the field experiments by the relative velocity on the blade which is sampled by a Pitot tube. This information has now been added to the manuscript.

  Regarding the LES, yes, as suggested the spectrum on a rotating blade section is different than that from the free-stream. However, as is now highlighted in the text, the rotational effects are not taken into account for the LES simulations for the following reasons: Firstly, the length scales of the added inflow turbulence are far lower than those in the free atmosphere on account of computational limitations on the size of

the domain. Secondly, it is of interest to study the response of the boundary layer to broad-band free-stream turbulence which includes lower frequencies not typically achieved in a wind tunnel test, but are present in the atmosphere. Thus for the purpose of the study, which is not a direct comparison to the experiment, but one to study the response of a blade section to inflow turbulence of the broad-band kind, the chosen methodology not taking into account rotation suffices.

Regarding the application of Mack's relation. This relation is the result of wind tunnel tests where rotation is not taken into account and also importantly, low-frequency disturbances are not the ones with the highest energy (in a wind tunnel) which is the case in the atmosphere (see Schaffarczyk et al., 2017 cited in the manuscript). Thus, the LES serves as a wind tunnel test including these missing lower frequency components, but does not take into account rotational effects.

- **Title says transition mechanisms on Multi MW wind turbine blades, but this can range from 2 to ∞.**
  That is true. Thus, we changed the title from Multi-Megawatt to Megawatt.

- **Addition of rated power in Table 1**
  Rated powers have been added to Table 1.

- **Table 1: Information about airfoils involved**
  Airfoil types are now included in Table 1 when applicable.

- **Inclusion of the field experiment by O. Pires et al. (2018) in Table 1.**
  This reference has been included.

- **DanAERO: More information about the experiment, specifically information about microphones and pressure taps**
  The explanation is added to the article.

- **Making Fig. 1 color-blind friendly and have extended explanation**
  Figure 1 is updated according to a color-blind friendly color palette. The T-S wave peaks and laminar and turbulent spectra are shown in the figure and further additional explanations have been added to the text according to reviewers' comments.

- **Make Fig. 3 color-blind friendly with distinguished lines**
  Figure 3 is updated according to color-blind friendly color palette and the lines and dots are made more distinct.

- **The legends of Figure 6 $x/c_{tr}$**
  The captions of the Figure 6 are changed according to the reviewers' comments. Thanks for noticing this.

- **Aerodynamic Glove experiment: Enhanced or replaced by thermography?**
  Results from the microphones are now also added to the text. The results from the microphones are limited in the sense that no large regions of laminar flow were seen during steady-state operation on account of the surface of the aerodynamic glove. Details have been added to the text and comparisons between transition prediction from

the microphones and thermography have been incorporated as well. This comparison also clarifies the other point made about equal accuracy between the two techniques.

- **Aerodynamic Glove experiment: Description of the measurement technique**
A description of the measurement techniques has been added. The distance does play a role in the final resolution that is possible, details of which have been included in the text. The thermography images are instantaneous and not averaged in time. This issue has been made clear in the text. The two teams did indeed use different detector materials, but two teams were asked if they could be a part of the project and both agreed. This is the reason why results from two teams are presented.

- **The text suggests that thermography is of equal value as microphones for transition detection.**
This was not the intended message. Transition can however be reliably detected using thermography and finds its applications in cases where a non-intrusive method is sought. The clarification of microphones being superior for the purpose of transition research has been made clear.

- **Section 3.1: Overall difficult to follow**
This section has been heavily edited to make it easier to follow with the inclusion of this section only to show the different RANS-based techniques that are presently used for wind turbine transition calculation for the sake of completeness to this article on wind turbine transition.

  In particular, the following points have been taken into account:

  - The title "Findings from IEA Wind TCP Task 29 Subection 3.6" has been revised. The reference is also made to Task 3.6 described in Chapter 10 instead of the entire study.
  - Yes, you are right, Table 2 refers to the meshes of the DAN-AERO blade, this has been made clear.
  - Table 3: $wt_{perf}$ is no longer referred to as an outdated BEM code.
  - It has been added that not just $c_p$ but also $c_T$ is underpredicted.
  - Note that the other points made regarding Section 3.1 no longer apply after the changes to the manuscript and are thus not addressed individually.

- **Line 227: URANS full rotor simulations for DAN-AERO experiments: About T.I. values calculated from experiments to apply in simulations**
The turbulence is quantified by the turbulence intensity ($T.I.$), which is the standard deviation of the relative velocity divided by the average relative velocity over 10 minutes in this case. Ten minutes average of the velocity data obtained from the Pitot tube on the blade is used in order to obtain the $T.I.$ values.

- **Mention the relative radial position instead of the absolute y value**
The relative radial positions are added in the manuscript

- **Figure 13: Indicate the distance of the blade section from the hub as done in Fig. 12**
  This has been done.

- **An estimate for changes in wind speed, turbulence intensity and angle of attack that the blade at section y = 36.8 undergoes**
  A text is added as follows: ¨The selected case is operating in low shear conditions. The angle of attack changes from 4 to 8 degrees, and the relative velocity on the blade changes from 62.5 to 66 m/s, showing the same trend with azimuth angle as the transition position in Figure 14. Furthermore, the inflow turbulence signals detected from a leading edge microphone show higher values (100 - 115 dB) between 0 to 150 degrees azimuth and decrease to 85 - 105 dB range between 150 to 350 degrees azimuth, showing an opposite response than other parameters with azimuth angle.¨
  In addition we are adding a supplementary plot to the response, presented below:

[Figure]

Figure 1:

- **Figure 14 adding that it is pressure side**
  This change has been made.

- **Line 271: Deriving aoa from force measurements**
  It is calculated from HAWC2 and an explanation is introduced in the text highlighted in red.

- **Section 3.3: LES on a non-rotating blade section**
  This is right, the LES has been performed on a non-rotating blade section and the text has been revised to make this very clear. The reasoning for doing this has also been elaborated.

- **Mention the relative thickness of airfoils**
  Added to the other parts of the manuscript.

- **Section 3.3: LES level of detail**
  You are right and the level of detail here was a lot higher. It has been cut down and references are made where necessary.

- **STIG: What does it mean?**
  STIG refers to Synthetic turbulence inflow generator. The abbreviation has been added on its first mention in the text.

- **Additional minor comments to the LES section**
  The following changes have been made:

  - The airfoil name corresponding to the plots has been included.
  - The label in the spanwise direction has been added.
  - The legend "up" on Fig. 18 (first version of the manuscript) refers to fluctuating velocity up(rime): $u'$. This has been changed to $w'$ since the inflow velocity direction across the manuscript has been changed to "$w$".
  - Yes, lower frequencies of the inflow turbulence on line 398. This has been made clear.
  - Klebanoff modes or boundary layer streaks which is the boundary layer response to external disturbances. A relevant citation has been included.

- **No evidence for laminar flow from mean pressure distributions**
  This is correct and none of the experiments discussed uses pressure distributions to detect laminar flow. The conclusion has been edited accordingly.

- **Line 500: On the suction side?**
  This is correct and has been made clear.

- **Line 502: Much less fully turbulent models could lead to a smaller deviation?**
  The transition location using transition models, especially correlation based transition models depends heavily on the inflow conditions and grid. To expect similar results across different codes would in the first place require the grids to be identical which is difficult to achieve.

We gratefully acknowledge the effort of the referee and his/her contributions in enhancing the quality of our paper. Thanks a lot.

B.A. Lobo, O.S. Özçakmak, H.A. Madsen, A.P. Schaffarczyk, M. Breuer, N.N. Sørensen

**Review # 2**

We appreciate the effort of Ryan Scott for evaluating our manuscript in detail. In the following his remarks are answered and modifications resulting from his comments are explained. Note that in the annotated version of the manuscript all modifications (replacements, additions and deletions) regarding the remarks of reviewer # 2 are highlighted in blue.

Please note that in response to both reviewers regarding a consistent writing style, the text was in general edited to achieve this by a single author. No highlights are made to such changes as the message delivered and information contained remains unaltered.

**Response to specific comments:**

- **Consistent writing style**
  As described above, the text has been revised for a consistent writing style.

- **Figure formatting varies between sections**
  The figures have been edited and all graphs are now plotted using the same tool when applicable.

- **Transitory Paragraphs**
  The text has been revised to transition more smoothly between the main sections better describing what to expect. Furthermore, methodologies have been added.

- **Differences in turbulence produced by an upstream turbine and synthetic turbulence and how this might impact the PSD plots**
  The first paragraph of Section 3.3 has been edited to elaborate on the kind of results one could expect from the LES which uses synthetic turbulence in the sense that it is similar to a wind tunnel but also includes lower frequencies that are absent when using active grids for the generation of turbulence in a wind tunnel. Furthermore, on account of the length scales in the free-atmosphere being different to that which is feasible using simulations, and furthermore on account of rotational effects a direct comparison is not possible between experiment and simulation.

  However, as already discussed in the conclusions of the LES section, the PSD plots from the simulations and experiments do have their share of similarities with natural transition in both instances being very obvious while in the presence of elevated turbulence intensity it becomes difficult to distinguish what is laminar and what is turbulent.

- **Why some quantities are represented i.e. vorticity ?**
  An explanation for this choice is added to the text in Section 3.2 in blue color.

- **Are x/c in Figure 5 and $X_{tr}$ in Figure 13 the same? Where is $X_{tr}$ defined?**
  The legends of Figure 13 have been updated to clarify this.

Other Points:

- Possibly place Figure 4 ahead of Figure 2.

  Figure 2 is moved ahead of Figure 4 as suggested by the reviewer.

We gratefully acknowledge the effort of the referee and his contributions in enhancing the quality of our paper. Thanks a lot.

B.A. Lobo, O.S. Özçakmak, H.A. Madsen, A.P. Schaffarczyk, M. Breuer, N.N. Sørensen

---

## Author Response (AR2)

Response to the Review on the Paper wes-2022-42

**On the laminar-turbulent transition mechanism on Multi-Megawatt wind turbine blades operating in atmospheric flow**

B.A. Lobo, O.S. Özçakmak, H.A. Madsen, A.P. Schaffarczyk, M. Breuer, N.N. Sørensen

**Review**

We appreciate the effort of Ryan Scott and Gerard Schepers for once again evaluating our manuscript in detail. In the following the remarks from Gerard Schepers are answered and modifications resulting from his comments are explained. Note that in the annotated version of the manuscript all modifications (replacements, additions and deletions) regarding the remarks are highlighted in red.

**Response to specific comments:**

- **Add a section: Conclusions and recommendations**
  Thanks for the suggestion. The last paragraph was intended to serve as the conclusion as you rightly pointed out. But it has now been enhanced with some more information which includes a few more conclusions as well as recommendations for future work as a new section.

- **Different font for $\mathcal{LESOCC}$**
  Yes, this is done on purpose as the code is usually referenced to using this font style. It is a typographical logo similar to LaTeX, for example.

We gratefully acknowledge the effort of the referees and their contributions in enhancing the quality of our paper. Thanks a lot.

B.A. Lobo, O.S. Özçakmak, H.A. Madsen, A.P. Schaffarczyk, M. Breuer, N.N. Sørensen